# Malaria oocysts require circumsporozoite protein to evade mosquito immunity

Feng Zhu [1,2,6], Hong Zheng [1,3,6], Suilin Chen [1,2], Kun Zhang [1,2], Xin Qin [1,2], Jingru Zhang [1,2], Taiping liu [1,2], Yongling Fan [1,2], Liting Wang [4], Xiaoxu Li [2,5], Jian Zhang [1,2 ✉] & Wenyue Xu [1,2 ✉]

Malaria parasites are less vulnerable to mosquito immune responses once ookinetes transform into oocysts, facilitating parasite development in the mosquito. However, the underlying mechanisms of oocyst resistance to mosquito defenses remain unclear. Here, we show that circumsporozoite protein (CSP) is required for rodent malaria oocysts to avoid mosquito defenses. Mosquito infection with $CSP_{mut}$ parasites (mutation in the CSP pexel I/II domains) induces nicotinamide adenine dinucleotide phosphate (NADPH) oxidase 5 (NOX5)-mediated hemocyte nitration, thus activating Toll pathway and melanization of mature oocysts, upregulating hemocyte TEP1 expression, and causing defects in the release of sporozoites from oocysts. The pre-infection of mosquitoes with the $CSP_{mut}$ parasites reduces the burden of infection when re-challenged with $CSP_{wt}$ parasites by inducing hemocyte nitration. Thus, we demonstrate why oocysts are invisible to mosquito immunity and reveal an unknown role of CSP in the immune evasion of oocysts, indicating it as a potential target to block malaria transmission.

[1] Department of Pathogenic Biology, Army Medical University (Third Military Medical University), Chongqing 400038, China. [2] Key Laboratory of Extreme Environmental Medicine, Ministry of Education of China, Chongqing 400038, China. [3] Department of Thoracic Surgery, Xinqiao Hospital, Army Medical University (Third Military Medical University), Chongqing 400038, China. [4] Biomedical Analysis Center, Army Medical University (Third Military Medical University), Chongqing 400037, China. [5] Department of High Altitude Physiology and Pathology, College of High Altitude Military Medicine, Army Medical University (Third Military Medical University), Chongqing 400038, China. [6] These authors contributed equally: Feng Zhu, Hong Zheng. ✉email: zhangjian@tmmu.edu.cn; xuwenyue@tmmu.edu.cn

Malaria, one of the most devastating diseases worldwide, is transmitted through the bite of a *Plasmodium*-infected mosquito. The development of malaria parasites in the *Anopheles* mosquito begins with the transformation of ingested gametocytes into gametes in the midgut. Microgametes fertilize macrogametes to generate zygotes, which are then activated to form motile ookinetes that transverse the peritrophic matrix and midgut epithelium. Underneath the midgut epithelium, zygotes transform into oocysts that generate thousands of sporozoites, which are then released into the mosquito hemolymph, and finally invade the salivary glands.

After feeding on a rodent host, blood taken into the mosquito midgut triggers the secretion of peroxidase/oxidase that catalyzes the crosslinking of proteins in the mucin layer of the mosquito midgut, which prevents gut bacteria and *Plasmodium* parasites from coming into direct contact with the midgut epithelium and eliciting an immune response[1]. However, a robust immune response is believed to be initiated when ookinetes begin to penetrate midgut epithelial cells. Ookinete invasion first induces the expression of nitric oxide synthase (NOS) and the production of nitric oxide (NO) in midgut epithelial cells[2] and then upregulates the expression of both heme peroxidase (HPX2) and nicotinamide adenine dinucleotide phosphate (NADPH) oxidase 5 (NOX5). These oxidases, together with NOS, mediate epithelial nitration, which make ookinetes visible to the mosquito immune system[3,4]. The nitrated midgut epithelial cells trigger the associated hemocytes to release hemocyte-derived microvesicles, which then activates complement-like thioester containing protein 1 (TEP1) to lyse parasites[5]. In addition to eliciting a local immune response, ookinetes can trigger systemic mosquito immune responses by inducing the midgut epithelium to secrete prostaglandin[6], which in turn, stimulates the production of hemocyte differentiation factor, which induces hemocyte differentiation and establishes immune memory against the parasites[7,8]. To facilitate their development in mosquitoes, ookinetes have developed strategies to escape mosquito immune responses. For instance, *Pfs47*, a *Plasmodium falciparum* gene, has been shown to suppress the mosquito immune system, allowing the parasite to infect *Anopheles gambiae*[9,10]. *Pfs47* disrupts JNK-mediated apoptosis and suppresses nitration of ookinete-invaded midgut cells, which are critical for activating the complement-like system[4,11].

The mosquito immune response can also limit the development of oocysts through the generation of NOS dependent on a signal transducer and activator of transcription (STAT)[12], hemocyte differentiation induced by lipopolysaccharide-induced TNFα transcription factor-like transcription factor 3[12,13], and pro-phenoloxidase (PPO) produced by hemocytes[14,15]. However, as compared to ookinetes, oocysts are less susceptible to the effects of mosquito immune defense, which facilitates their survival and development in mosquito hosts[16]. In addition, the lowest number of parasites in mosquitoes occurs at the oocyst stage, strongly suggesting that this stage is the most vulnerable for the disruption of malaria transmission[16]. Therefore, understanding oocyst immune evasion strategies could help toward designing strategies to block malaria transmission[16].

In this work, we find that a mutation of the circumsporozoite protein (CSP) pexel I/II domain significantly alters the conformation of this protein, which make the parasites visible to the mosquito immune system, leading to the nitration of hemocytes, activation of the Toll pathway, and the release of TEP1. As a result, the release of $CSP_{mut}$ (CSP pexel I/II domain mutant parasite) sporozoites is greatly impaired due to TEP1-mediated melanization of mature oocysts. Thus, we reveal an unknown role of CSP in evading mosquito immune responses and suggest CSP as a potential target to block malaria transmission.

## Results

### The release of sporozoites from $CSP_{mut}$ oocyst in mosquitoes is greatly impaired.

The pexel I/II domain of CSP has been demonstrated to mediate its translocation from the parasitophorous vacuole into the cytoplasm of hepatocytes[17]; however, whether this domain influences the development of malaria parasites in mosquitoes is still unknown. A recombinant malaria parasite of rodents, *Plasmodium yoelii* 265 BY $CSP_{mut}$, was successfully generated by replacing the wild-type *CSP* with a *CSP* pexel I/II mutant using CRISPR-Cas9 technology (Fig. 1a, and Supplementary Fig. 1). The first, third, and fifth amino acids of pexel I (RNLNE) and pexel II (RLLGD) in wild-type CSP presented mutations from R, L, E and R, L, D to A, as previously reported[18]. Both the parasitemia and gametocytemia of mice infected with *P. yoelii* $CSP_{mut}$ were comparable to those of mice infected with the wild-type parasite (*P. yoelii* $CSP_{wt}$), indicating that the CSP pexel I/II mutation had no effect on the development of either the asexual or sexual blood stages of the malaria parasite (Fig. 1b, c). In addition, both the number and size of $CSP_{mut}$ oocysts were comparable to those of $CSP_{wt}$ parasites at the indicated time points in mosquitoes (Fig. 1d, e). Although a few sporozoites were observed in the hemolymph, none were found in the salivary glands of mosquitoes infected with *P. yoelii* $CSP_{mut}$ (Fig. 1f, g). We also successfully constructed a *P. berghei* $CSP_{mut}$ mutant[19], and similar results were obtained (Fig. 1h, i), implying that there was no species-specific effect on parasite development in mosquitoes when the CSP pexel I/II domain was mutated. Thus, our results strongly suggest that the release of sporozoites from $CSP_{mut}$ oocysts was significantly impaired.

### TEP1-mediated oocyst melanization contributes to the developmental arrest of mutant parasites.

To elucidate the mechanism underlying the developmental arrest, the development of *P. yoelii* $CSP_{mut}$ oocysts was investigated in detail. On day 9 post infection (PI), we found that melanization of mature $CSP_{mut}$ oocysts began, but with different amounts of black dots deposited inside the oocyst capsule on day 12 PI (Fig. 2a). Under electronic microscopy, some mature $CSP_{mut}$ oocysts generated thousands of sporozoites with normal morphology, but many sporozoites were found to be vacuolated and partially melanized at day 11 PI. This was not observed for the matured $CSP_{wt}$ oocysts (Fig. 2b). In addition, the percentage of the melanized $CSP_{mut}$ oocysts increased from an average of 28.2% on day 9, to 49.3% on day 12, and 73.2% on day 15 PI. No melanized oocysts were observed in $CSP_{wt}$-infected mosquitoes at any of the time points (Fig. 2c). To confirm the decrease in the number of released sporozoites from melanized mutant oocysts, midguts from both $CSP_{wt}$ and $CSP_{mut}$ parasite-infected mosquitoes were isolated on day 10 PI, incubated in vitro, and their capacity to release sporozoites was then compared. Mature $CSP_{mut}$ oocysts released considerably fewer sporozoites than mature $CSP_{wt}$ oocysts when incubated for longer than 2 h (Fig. 2d). Thus, the melanization of mature oocysts was closely associated with the impaired release of sporozoites from $CSP_{mut}$ oocysts.

In mosquitoes, melanization characterized by the synthesis and deposition of melanin was the response to remove dead parasites. Since TEP1 plays an important role in the killing of malaria parasites[20–24], we investigated whether TEP1 was involved in the melanization of $CSP_{mut}$ oocysts. Both the expression and activation of TEP1 were evaluated in mosquitoes infected with $CSP_{mut}$ and $CSP_{wt}$ parasites. *TEP1* mRNA levels were significantly upregulated in mosquitoes infected with the mutant parasites at day 5 PI, as compared to that observed for the $CSP_{wt}$ infection (Fig. 2e). Consistently, more TEP1 was activated in the hemolymph of mosquitoes infected with mutant parasites than

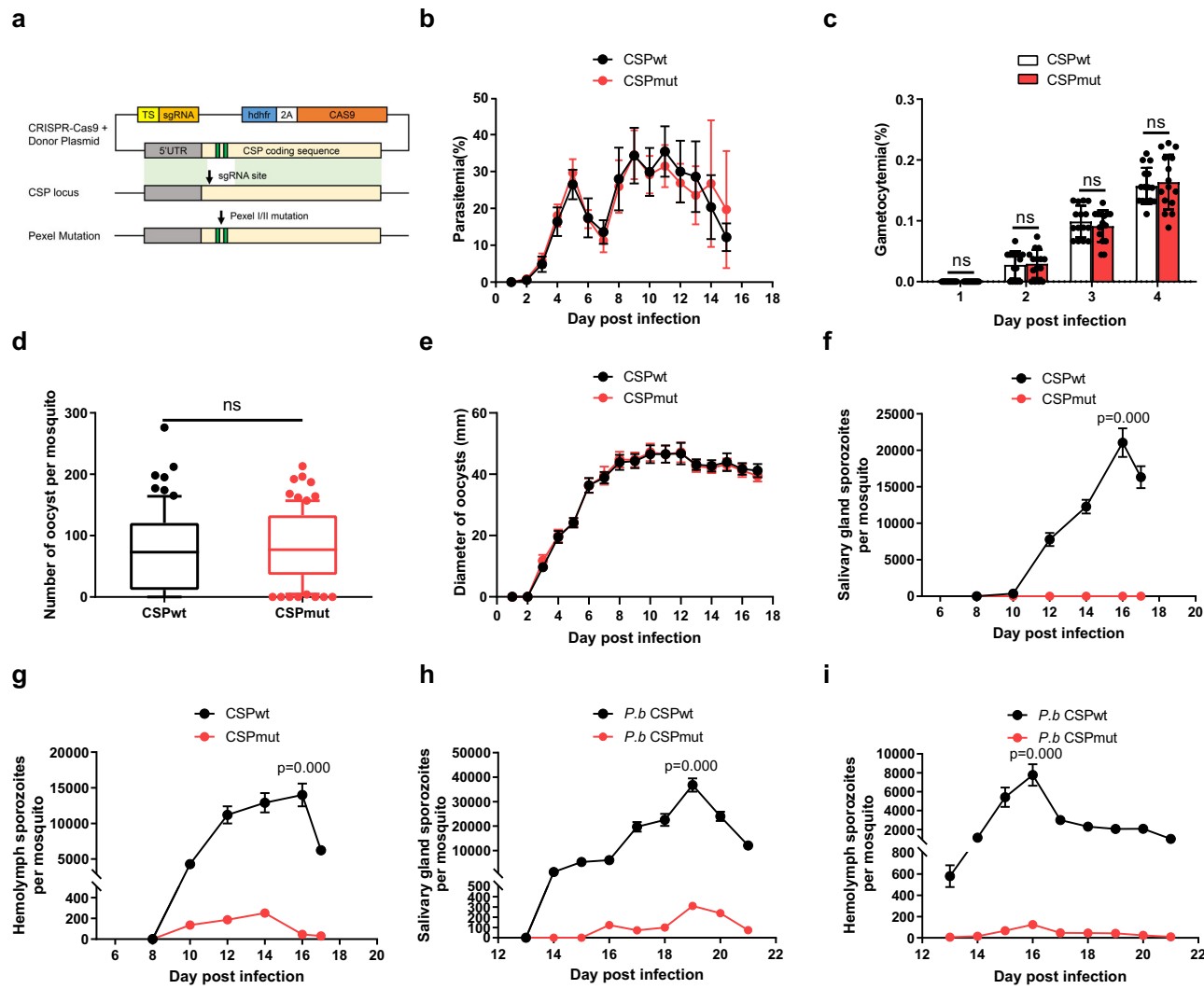

**Fig. 1 Release of sporozoites from CSP_mut oocyst in mosquitoes is greatly impaired. a** Schematic representing the construction of the recombinant *Plasmodium yoelii* CSP_mut parasite using CRISPR-Cas9. The amino acids of pexel I (RNLNE) and pexel II (RLLGD) in wild-type CSP were mutated into ANANA and ALAGA, respectively. Both the parasitemia (**b**) and gametocytemia (**c**) of mice (*n* = 15) infected with the CSP_wt or CSP_mut parasites were compared at the indicated time points. The average number of oocysts (**d**) in mosquitoes (*n* = 81) at day 7, and the average diameter of oocysts (**e**) under ×400 field in mosquitoes (*n* = 40) at the indicated time points after feeding on CSP_wt or CSP_mut-infected mice. The average number of salivary gland sporozoites (**f**) in mosquitoes (*n* = 90), and hemolymph sporozoites (**g**) in mosquitoes (*n* = 60) infected with *P. yoelii* CSP_wt or CSP_mut parasites at the indicated time points were recorded. The average number of salivary gland sporozoites (**h**) in mosquitoes (*n* = 90), and hemolymph sporozoites (**i**) in mosquitoes (*n* = 60) infected with *P. berghei* CSP_wt or CSP_mut parasites at the indicated time points. For box-plot diagram (**d**): middle line represents median; boxes extend from the 25th to 75th percentiles. The whiskers mark the 10th and 90th percentiles. **c**, **d** A two-sided Student's *t*-test was used for the statistical analysis. **f–i** A two-sided Mann–Whitney *U* test was used for the statistical analysis. The data were presented as the means ± SD; ns, no significance; *p*-values were shown. The data were combined based on three independent experiments (**b–i**). Source data are provided as a Source Data file.

those infected with CSP_wt parasites at day 5 PI (Fig. 2f). Interestingly, TEP1 was only observed to bind to the surface of CSP_mut oocysts and co-localized with CSP protein inside the mutant oocysts, but not for CSP_wt oocysts at day 7 PI (Fig. 2g). When *TEP1* was silenced, however, no oocysts were found to be melanized in CSP_mut parasite-infected mosquitoes on day 12 PI (Fig. 2h), and an increased number of CSP_mut sporozoites appeared in the hemolymph (Fig. 2i). Both the motility and infectivity of CSP_mut hemolymph sporozoites were comparable to those of the CSP_wt hemolymph sporozoites in vitro (Supplementary Movie 1, 2 and Supplementary Fig. 2), indicating no biological defect in mutant parasites.

The melanization of the killed parasites was initiated by melanin biosynthesis, which is mediated by PPO cascades[25]. In addition, only hemocyte-derived PPOs, such as PPO1, PPO2,

PPO3, and PPO9, have been reported to mediate oocyst killing[15,26]. Therefore, we next investigated whether those PPOs were involved in this process. As a result, we found that *TEP1* knockdown could only significantly downregulate the mRNA levels of *PPO2*, *PPO3*, and *PPO9*, but not *PPO1*, in mosquitoes infected with the CSP_mut parasite (Supplementary Fig. 3), indicating the involvement of PPOs in the melanization of oocysts. Thus, the results demonstrated that the arrested development of CSP_mut parasites was attributed to the TEP1-mediated killing of oocysts, which might involve PPOs.

**CSP_mut infection induces hemocytes to release TEP1 in a Toll pathway-dependent manner.** We also found that more hemocytes were recruited to the midgut of mosquitoes infected with mutant parasites than those infected with WT parasites (Fig. 3a).

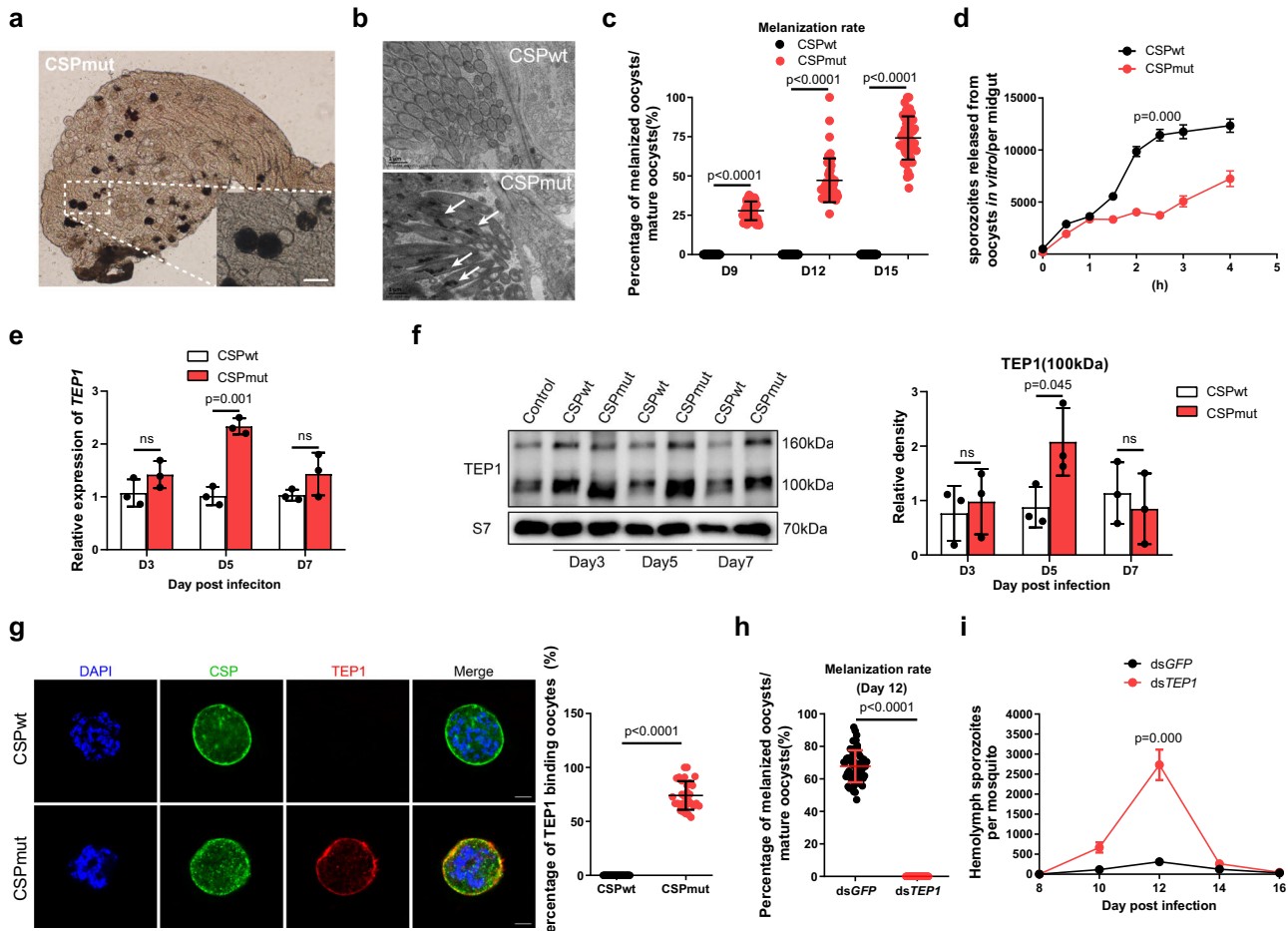

**Fig. 2 Parasite development arrest largely depends on TEP1-mediated mature CSP$_{mut}$ oocyst melanization. a** Representative image of melanized *Plasmodium yoelii* CSP$_{mut}$ in mosquitoes under a light microscope on day 12 after a blood meal; scale bar 50 μm. **b** Representative electron micrograph of vacuolized or melanized sporozoites (white arrows) from *P. yoelii* CSP$_{wt}$ and CSP$_{mut}$ oocysts at day 11; scale bar 1 μm. **c** Statistical analysis between the percentage of melanized mature *P. yoelii* CSP$_{wt}$ and CSP$_{mut}$ oocysts in mosquitoes ($n = 45$) at the indicated time points post blood feeding. **d** Average number of sporozoites released from *P. yoelii* CSP$_{wt}$ and CSP$_{mut}$ oocysts in mosquitoes ($n = 60$) at indicated incubation times in vitro. **e** mRNA expression level of *TEP1* in mosquitoes ($n = 15$) infected with *P. yoelii* CSP$_{wt}$ and CSP$_{mut}$ at the indicated time points post blood feeding determined using real-time PCR. **f** TEP1 activation (100 kD band) in hemocytes from mosquitoes ($n = 40$) infected with *P. yoelii* CSP$_{wt}$ and CSP$_{mut}$ parasites detected using western blot; S7 protein was used as an internal control (left). Pooled data was quantified and analyzed (right). **g** Representative image of TEP1 binding to *P. yoelii* CSP$_{wt}$ and CSP$_{mut}$ oocysts at day 7 PI stained with anti-CSP repeat and anti-TEP1 (left), and the percentage of oocysts from *P. yoelii* CSP$_{wt}$ or CSP$_{mut}$ parasite-infected mosquitoes ($n = 30$) bound with TEP1 was statistically analyzed (right); scale bar 10 μm. Percentage of melanized mature oocysts (**h**) in *P. yoelii* CSP$_{mut}$ parasite-infected mosquitoes ($n = 60$) at day 12 PI, and average number of hemolymph sporozoites (**i**) in mosquitoes ($n = 60$) infected with CSP$_{mut}$ parasites at the indicated time points compared following *TEP1* knockdown. **d–f** A two-sided Student's *t*-test was used for the statistical analysis. **c, g–i** A two-sided Mann–Whitney *U* test was used for the statistical analysis. The data were presented as the means ± SD; ns, no significance; *p*-values were shown. The data were combined based on three independent experiments (**c–i**) or two independent experiments (**b**). Source data are provided as a Source Data file.

As cellular immune responses play an essential role in mosquito immunity[5], the results further confirmed the ability of the mutant parasites to trigger mosquito immune defenses. Although a previous study argued that TEP1 was predominantly produced in the fat body[27], more evidence has indicated hemocytes as the main source of TEP1[26,28]. Therefore, we examined TEP1 expression in all three types of mosquito hemocytes: prohemocytes, granulocytes, and oenocytoids. No significant TEP1 expression was detected in prohemocytes from mosquitoes infected with either CSP$_{mut}$ or CSP$_{wt}$ parasites. However, as compared to CSP$_{wt}$ parasite-infected mosquitoes, a higher ratio of granulocytes from CSP$_{mut}$ parasite-infected mosquitoes expressed TEP1, as determined by either morphology identification (Fig. 3b) or flow cytometry analysis of granulocytes (Supplementary Fig. 4). Chemical depletion of granulocytes using clodronate liposomes

(CLDs) (Supplementary Fig. 5), as previously reported[26], decreased both *TEP1* expression and the percentage of melanized CSP$_{mut}$ oocysts but increased the number of hemolymph sporozoites (Fig. 3c). Although flow cytometry could not accurately distinguish the TEP1-expressed oenocytoids from granulocytes (Supplementary Fig. 4), morphological identification suggested the upregulation of TEP1 in oenocytoids from CSP$_{mut}$ parasite-infected mosquitoes (Fig. 3b). Thus, we demonstrated the essential role of hemocyte-derived TEP1 in the melanization of CSP$_{mut}$ oocysts.

Next, we investigated the underlying mechanism of enhanced TEP1 expression in hemocytes induced by CSP$_{mut}$ parasite infection. mRNA isolated from CSP$_{wt}$ and CSP$_{mut}$ parasite-infected mosquitoes at day 7 PI, 2 days before CSP$_{mut}$ oocyst melanization, was observed, sequenced, and compared

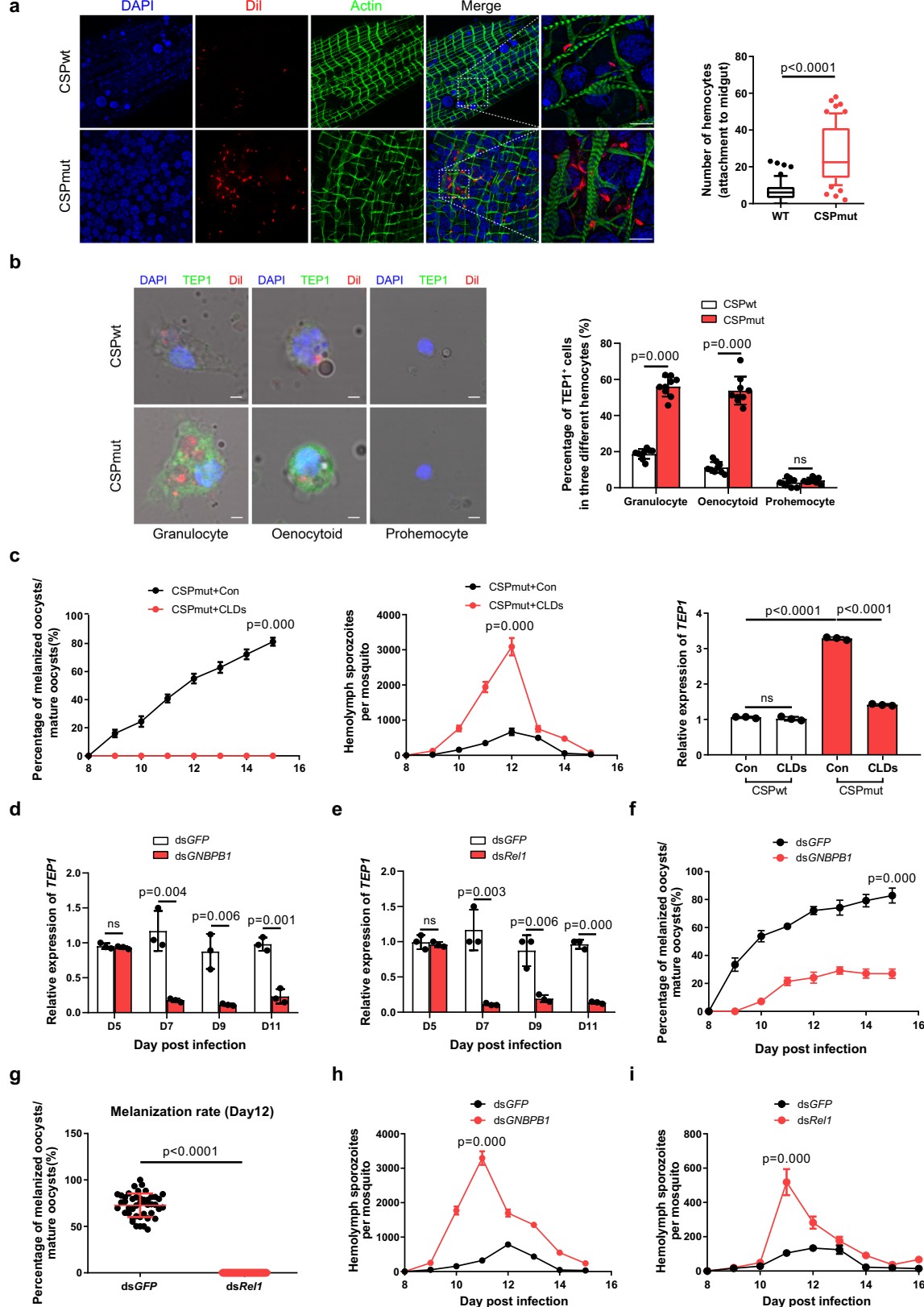

(Supplementary Table 1). As TEP1 is reportedly regulated by the Toll, immune deficiency (IMD), and Janus kinase–STAT signaling pathways[23], genes in each pathway were selected for comparison between CSP$_{wt}$ and CSP$_{mut}$ parasite-infected mosquitoes. No significant difference in the expression of *IMD* (ASTE010360), *JNK* (ASTE007551) and *STAT* (ASTE011642)

involved in the IMD, JNK, and STAT signaling pathways was found between CSP$_{wt}$- and CSP$_{mut}$-infected mosquitoes. However, those belonging to the Toll signaling pathway, such as *Gram-negative binding protein B1* (*GNBP-B1*) (ASTE016199) and *Relish-like protein 1* (*Rel1*) (ASTE011378), were significantly upregulated (Supplementary Fig. 6a), which was further validated

**Fig. 3 Infection with CSP$_{mut}$ induces hemocytes to release TEP1 in a Toll-dependent manner. a** The recruitment of hemocytes, which were stained with a hemocyte-specific lipophilic dye (Vybrant CM-DiI), to the midgut of mosquitoes at day 7 after infection with *Plasmodium yoelii* CSP$_{wt}$ or CSP$_{mut}$ parasites (left), and statistical analysis of hemocytes recruited to the midguts ($n = 64$) (right); scale bar 50 μm. **b** A representative immunofluorescent confocal image of granulocyte, oenocytoid, and prohemocyte from *P. yoelii* CSP$_{wt}$ or CSP$_{mut}$ parasite-infected mosquitoes ($n = 60$) at day 5 PI, stained with anti-TEP1 antibody (left), the percentage of TEP1 positive cells in granulocyte, oenocytoid, and prohemocyte was quantified (right). Scale bar 2 μm. **c** The percentage of melanized mature oocysts (left) in mosquitoes ($n = 45$), the average number of hemolymph sporozoites (middle) in mosquitoes ($n = 45$), and the expression of TEP1 (right) in mosquitoes ($n = 45$) infected with *P. yoelii* CSP$_{wt}$ and CSP$_{mut}$, pretreated with or without clodronate liposome. The mRNA levels of *TEP1* in CSP$_{mut}$ parasite-infected mosquitoes ($n = 15$) determined using real-time PCR, after *GNBP-B1* (**d**) or *Rel1* (**e**) knockdown. The percentage of melanized mature oocyst in CSP$_{mut}$ parasite-infected mosquitoes ($n = 45$) determined after knockdown of *GNBP-B1* (**f**) or *Rel1* (**g**). The number of hemolymph sporozoites in CSP$_{mut}$ parasite-infected mosquitoes ($n = 45$) counted after knockdown of *GNBP-B1* (**h**) or *Rel1* (**i**). For box-plot diagram (**a**): middle line represents median; boxes extend from the 25th to 75th percentiles. The whiskers mark the 10th and 90th percentiles. **b**, **d**–**f** A two-sided Student's *t*-test was used for the statistical analysis. **a**, **c**, **g**–**i** A two-sided Mann–Whitney *U* test was used for the statistical analysis. The data were presented as the means ± SD; ns, no significance; *p*-values were shown. The data were combined based on three independent experiments (**a**–**i**). Source data are provided as a Source Data file.

by quantitative real-time PCR (Supplementary Fig. 6b). Their roles in hemocyte TEP1 regulation during infection with CSP$_{mut}$ parasites were determined by RNA silencing. As a result, the inhibition of either JNK, STAT, or IMD signaling pathways by knockdown of the related genes, *JNK* (ASTE007551), *STAT* (ASTE011642), or *IMD* (ASTE010360), respectively, had no significant effect on the melanization of CSP$_{mut}$ oocysts (Supplementary Table 2). However, knockdown of Toll pathway-related genes, *GNBP-B1* and *Rel1*, significantly reduced *TEP1* expression (Fig. 3d, e), and inhibited CSP$_{mut}$ oocyst melanization (Fig. 3f, g) but significantly increased the number of hemolymph sporozoites (Fig. 3h, i). Thus, we demonstrated that infection with CSP$_{mut}$ parasites significantly induced *TEP1* expression in a Toll pathway-dependent manner, which is closely associated with the melanization of mature mutant oocysts.

**CSP$_{mut}$ parasites trigger mosquito immune responses through the induction of hemocyte nitration.** Next, we investigated the underlying mechanism behind Toll pathway activation by CSP$_{mut}$ infection. *Wolbachia* infection in mosquitoes activates the Toll pathway by inducing the production of reactive oxygen species in *Aedes aegypti*;[29] therefore, we selected antioxidant genes for analysis in mosquitoes infected with CSP$_{wt}$ or CSP$_{mut}$ parasites. Importantly, genes encoding heme peroxidase, such as *HPX2*, *DBLOX* (*double heme peroxidase*), *NOX5*, and *DUOX* (*dual oxidase*), were significantly upregulated in CSP$_{mut}$-infected mosquitoes (Supplementary Fig. 7a, b). HPX2 and NOX5 are critical for the nitration of ookinete-invaded midgut epithelial cells and trigger the mosquito immune response against parasites[1]. This prompted us to investigate whether infection with CSP$_{mut}$ parasites could induce the nitration of recruited hemocytes and elicit an immune response. A significantly higher level of nitration was observed in hemocytes around the oocysts of mosquitoes infected with CSP$_{mut}$ parasites, compared to the CSP$_{wt}$ parasite-infected mosquitoes (Fig. 4a). Interestingly, some nitrated hemocytes were found to attach to the CSP$_{mut}$ oocysts (Supplementary Fig. 8) and simultaneously express TEP1 (Fig. 4a), indicating that some hemocytes recruited to mutant oocysts might be activated to express TEP1. To investigate the molecular mechanism underlying hemocyte nitration induced by CSP$_{mut}$ parasite infection, we silenced the upregulated antioxidant genes (*NOX5*, *DUOX*, *DBLOX*, and *HPX2*) and determined the effect on hemocyte nitration. Knockdown of *NOX5* abrogated the nitration of hemocytes in CSP$_{mut}$-infected mosquitoes (Fig. 4a) and resulted in the downregulation of *GNBP-B1*, *Rel1*, and *TEP1* mRNA levels, and the disappearance of CSP$_{mut}$ oocyst melanization, increased the number of sporozoites in the hemolymph (Fig. 4b–d), but the silencing of *DUOX* or *DBLOX* has no significant effect on the melanization of CSP$_{mut}$ oocysts (Supplementary Table 2). In

contrast to the important role that HPX2 plays in the nitration of ookinete-invaded midgut epithelial cells, knockdown of *HPX2* had no significant effect on hemocyte nitration, oocyst melanization, or the number of sporozoites in the hemolymph of CSP$_{mut}$-infected mosquitoes (Supplementary Fig. 9). Therefore, infection with mutant parasites results in the nitration of hemocytes and elicits a mosquito immune response to limit parasite development.

**CSP conformation is altered in CSP$_{mut}$ parasites and induces hemocyte nitration.** It has been reported that CSP expression levels in oocysts affect sporozoite morphology[30]. This prompted us to investigate whether the developmental defect observed in CSP$_{mut}$ parasites was due to the reduced CSP expression. The expression of CSP could be detected as early as day 4 PI in midgut oocysts using real-time PCR and detected at day 5 PI using western blot (Supplementary Fig. 10). However, the expression levels of CSP in CSP$_{wt}$ oocysts was comparable to that observed in CSP$_{mut}$ oocysts (Fig. 5a and Supplementary Fig. 10). Thus, the effect of CSP expression on the developmental defects of CSP$_{mut}$ parasites could be excluded. Evidence has shown that CSP undergoes a conformational change when sporozoites are released from the oocyst into the hemolymph. The thrombospondin repeat domain (TSR) at the C-terminus of CSP is exposed at the surface of oocyst sporozoites but is masked at hemolymph sporozoites surface by the N-terminus[31]. We therefore speculated that the mutation of the pexel I/II domain at the CSP N-terminus may result in a conformational change. Although anti-C-terminal CSP polyclonal antibody detected oocysts and hemolymph sporozoites of both the WT and mutant parasites (Fig. 5b), only mutant oocysts and hemolymph sporozoites were detected with anti-N-terminal CSP polyclonal antibody (Fig. 5b). Thus, our results strongly suggest that the mutation of the pexel I/II domain resulted in a conformational change of the CSP N-terminus. We noted a slightly different pattern in the WT hemolymph sporozoites labelled by anti-C-terminal CSP polyclonal antibody between ours and a previous study[31]. This discrepancy may be explained by the smaller peptide of the C-terminus of CSP used for the preparation of the anti-sera in our experiment, whereas the full length TSR region in the C-terminus of CSP was used in the previous study[31].

To test whether this CSP conformational change triggers the nitration of hemocytes recruited to midgut basal lamina, both WT and mutant oocysts were incubated with hemocytes from naïve mosquitoes, and hemocyte nitration was determined. A higher number of hemocytes were found to be nitrated when incubated with mutant oocysts than with CSP$_{wt}$ oocysts (Fig. 5c). Furthermore, much higher expression levels of *GNBP-B1*, *Rel1*, and *TEP1* were detected in hemocytes stimulated with CSP$_{mut}$

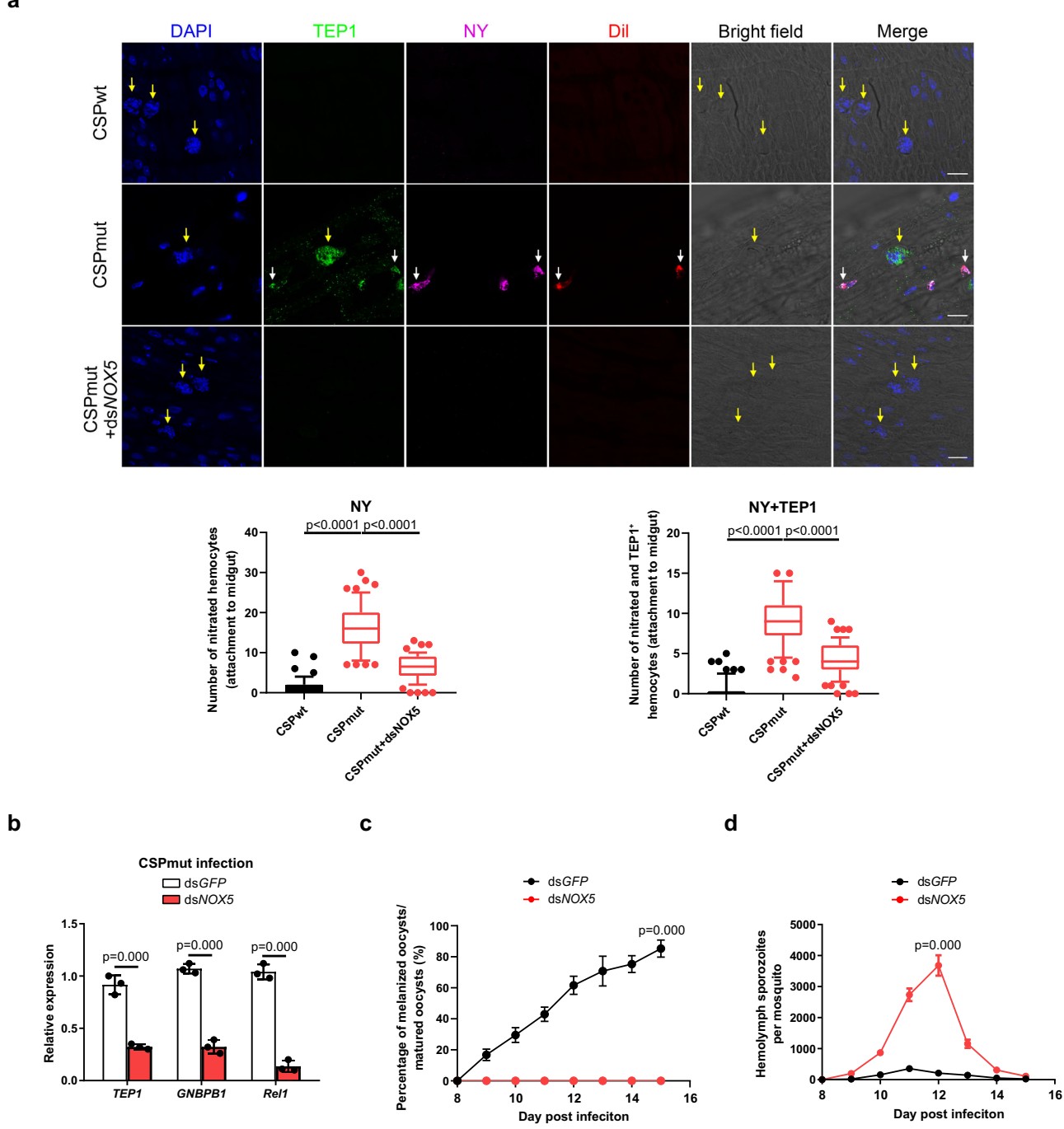

**Fig. 4 CSP$_{mut}$ triggers mosquito immune responses through the induction of hemocyte nitration. a** Representative image of the nitration and TEP1 expression of hemocytes around oocysts in *Plasmodium yoelii* CSP$_{wt}$ or CSP$_{mut}$ parasite-infected mosquitoes at day 4 after *NOX5* was knocked down or not. This was determined by the detection of nitrotyrosine (NY), TEP1, and Vybrant CM-DiI (DiI); scale bar 50 μm (upper), and the numbers of nitrated hemocytes and TEP1-positive nitrated hemocytes attached to midguts ($n = 60$) were statistically analyzed (low). Yellow arrow indicates either WT or mutant oocysts; white arrow indicates hemocytes. **b** The relative expression of *TEP1*, *GNBP-B1*, and *Rel1* in *P. yoelii* CSP$_{mut}$ parasite-infected mosquitoes ($n = 15$) at day 5 PI after *NOX5* knockdown. **c** The percentage of melanized mature oocyst in *P. yoelii* CSP$_{mut}$ parasite-infected mosquitoes ($n = 36$) at the indicated time points after *NOX5* knockdown. **d** The average number of hemolymph sporozoites in *P. yoelii* CSP$_{mut}$ parasite-infected mosquitoes ($n = 45$) at the indicated time points after *NOX5* knockdown. For box-plot diagram (**a**): middle line represents median; boxes extend from the 25th to 75th percentiles. The whiskers mark the 10th and 90th percentiles. **a**, **b** A two-sided Student's *t*-test was used for the statistical analysis. **c**, **d** A two-sided Mann–Whitney *U* test was used for the statistical analysis. The data were presented as the means ± SD; ns, no significance; *p*-values were shown. The data were combined based on three independent experiments (**a**–**d**). Source data are provided as a Source Data file.

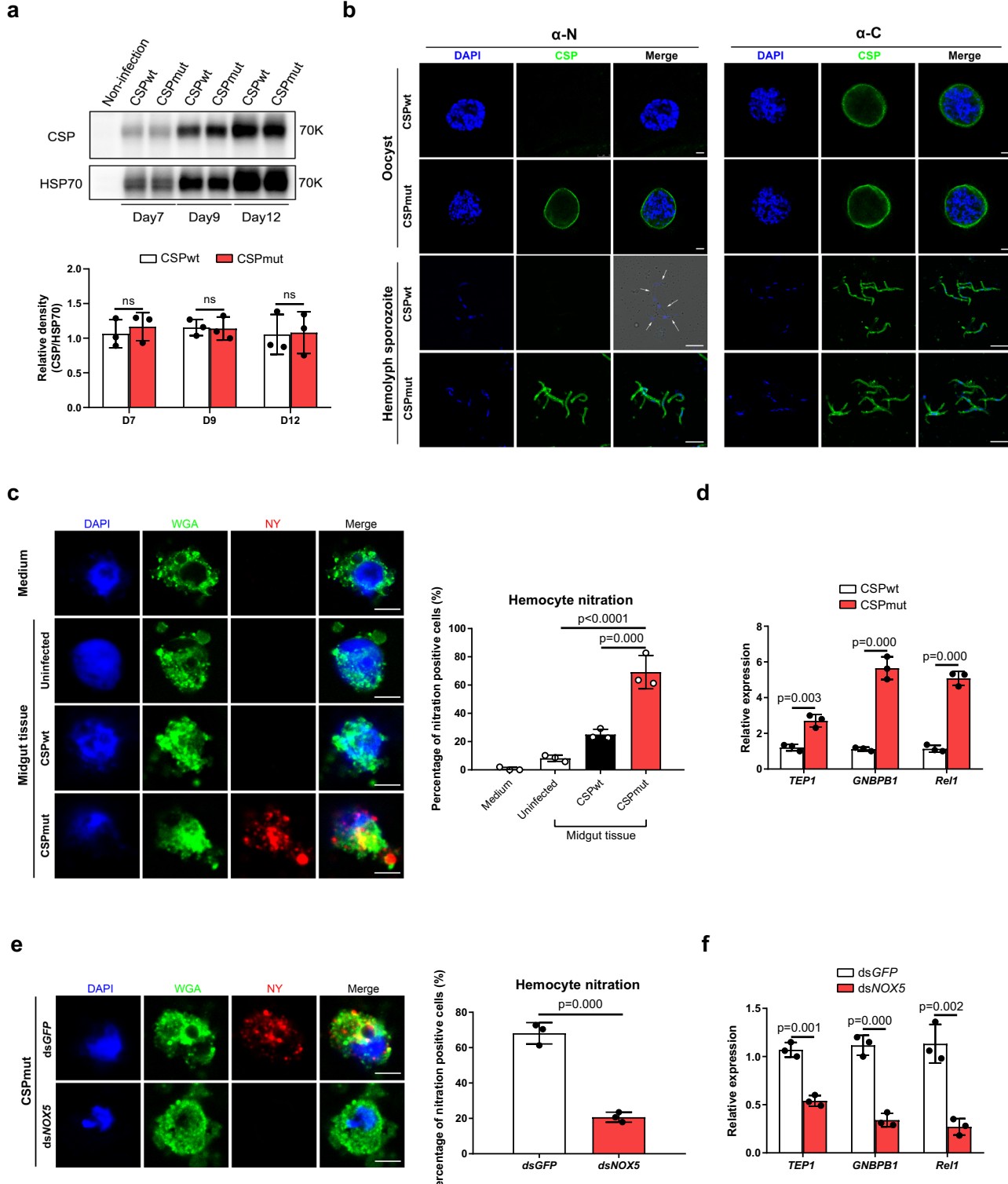

oocysts (Fig. 5d). However, nitration and enhanced expression of *GNBP-B1*, *Rel1*, and *TEP1* were abrogated in NOX5-silenced hemocytes when incubated with CSP$_{mut}$ oocysts (Fig. 5e, f). Thus, our findings demonstrated that the NOX5-mediated nitration of hemocytes and the mosquito immune response were elicited by the CSP conformational change in CSP$_{mut}$ parasites.

**Pre-infection of mosquitoes with CSP$_{mut}$ reduces parasite burden when challenged with RFP-expressing WT parasites**. To

further confirm that the mosquito immune response can be elicited by these mutant parasites, mosquitoes were first infected with CSP$_{wt}$ or CSP$_{mut}$ and then challenged with a red fluorescent protein (RFP)-expressing WT parasite (265BY *P. yoelii*-RFP) at day 3 PI. This time point was selected for the re-challenge because a robust mosquito immune response was stimulated by mutant oocyst CSP and mainly targeted oocysts of the re-challenged parasite. After the RFP-expressing WT parasite re-challenged at day 3 PI developed into early oocysts, the firstly infected mutant

**Fig. 5 CSP conformation of CSP_mut is altered and induces the nitration of hemocytes. a** Representative image of CSP relative expression in *Plasmodium yoelii* CSP_wt and CSP_mut oocysts from mosquitoes ($n = 40$) detected by western blot at days 7, 9, and 12 post blood meal (upper); CSP relative expression was statistically analyzed (lower). **b** Representative immunofluorescence image of oocyst (scale bar 10 μm) and hemolymph sporozoites (scale bar 5 μm) from *P. yoelii* CSP_wt or CSP_mut parasite-infected mosquitoes stained with anti-CSP-N and anti-CSP-C terminal antibodies. **c** Representative immunofluorescence image of in vitro nitration of hemocytes from naïve mosquitoes ($n = 20$) stimulated by midguts ($n = 20$) from *P. yoelii* CSP_wt or CSP_mut parasite-infected mosquitoes at day 5 PI (left). WGA was used to distinguish hemocytes, and hemocyte nitration was determined using anti-nitrotyrosine staining; scale bar 5 μm. Percentage of the nitrated hemocytes was statistically analyzed (right). **d** *GNBP-B1*, *Rel1*, and *TEP1* expression in hemocytes from naïve mosquitoes ($n = 20$) following incubation with midguts ($n = 20$) from *P. yoelii* CSP_wt or CSP_mut parasite-infected mosquitoes at day 5 PI determined using real-time PCR. **e** Representative immunofluorescence image of the nitration of hemocytes from naïve mosquitoes ($n = 20$) incubated with midguts from *P. yoelii* CSP_mut parasite-infected mosquitoes ($n = 20$) at day 5 PI with or without *NOX5* silenced (left); scale bar 5 μm. Hemocyte nitration was determined as described in **c** and statistically analyzed (right). **f** *GNBP-B1*, *Rel1*, and *TEP1* expression in hemocytes from naïve mosquitoes ($n = 20$) with or without ds*NOX5* incubated with midguts ($n = 20$) from *P. yoelii* CSP_mut parasite-infected mosquitoes at day 5 PI. A two-sided Student's *t*-test was used for the statistical analysis. The data were presented as the means ± SD; ns, no significance; *p*-values were shown. The data were combined based on three independent experiments (**a**, **c**–**f**) or two independent experiments (**b**). Source data are provided as a Source Data file.

parasites were developed for approximately 5 days. At that time, the developed mutant oocysts would trigger mosquito immune responses against the re-challenged parasites.

Although the number of 265BY *P. yoelii*-RFP oocysts was comparable between mosquitoes pre-infected with CSP_mut and CSP_wt parasites (Fig. 6a), oocyst melanization in pre-infected mosquitoes began at day 9 PI, and the percentage of melanized oocysts increased from 16% on day 9 to 67%, on day 15 PI (Fig. 6b). Here, it should be noted that the definitive origin of melanized oocysts was difficult to determine, as the fluorescence could be faded as the RFP-expressing oocyst was melanized. We observed the melanized RFP-expressing oocyst at both light and fluorescent microscopy and found some melanized oocyst could still be identified by residual fluorescence (Fig. 6b). However, some melanized RFP-expressing oocysts without residual fluorescence could have been missed. Therefore, the actual percentage of the melanized RFP-expressing oocysts should be higher than what we have reported here. As a result, fewer RFP-expressing hemolymph and salivary gland sporozoites were observed in mosquitoes pre-infected with CSP_mut, as compared to those pre-infected with the CSP_wt parasites (Fig. 6c, d). This indicates that mosquitoes pre-infected with CSP_mut parasites develop resistance to re-challenge with the RFP-expressing WT parasite.

To test whether hemocyte nitration contributed to the resistance observed in CSP_mut pre-infected mosquitoes, *NOX5* was silenced prior to infection with 265BY *P. yoelii*-RFP. Following *NOX5* knockdown, melanization of RFP-expressing WT oocysts disappeared, although no change in oocyst number was observed (Fig. 6e, f). In addition, the number of hemolymph sporozoites increased, and a few infectious salivary gland sporozoites appeared (Fig. 6g, h). In addition, *NOX5* knockdown significantly inhibited *TEP1* expression and downregulated the genes of the Toll pathway (Fig. 6i). Therefore, we demonstrated that CSP_mut infection triggered the mosquito immune defense by inducing nitration, and CSP might be explored as a potential target to block malaria transmission.

## Discussion

Compared to the great loss of parasites that occurs during ookinete penetration of the mosquito midgut epithelium, only a small proportion of parasites are lost during the oocyst stage[32]. This indicates that oocysts tend to evade mosquito immune defenses; however, the underlying mechanism is still unknown. In the present study, we demonstrated that CSP, on the inner membrane of oocysts and the surface of sporozoites[30], is required for the evasion of mosquito immune responses. The change in CSP conformation through mutation of the pexel I/II domain made the parasite visible to the mosquito immune system through the induction of hemocyte nitration.

Although previous studies have shown that the mosquito immune response can limit oocyst development[12–15,33], oocysts are less vulnerable than ookinetes to the mosquito immune system[16]. This indicates that oocysts have evolved strategies to escape mosquito immunity to facilitate their survival. The incorporation of mosquito proteins such as laminin, matrix metalloprotease 1 (MMP1), and lysozyme c-1 (LYSC-1) into the oocyst capsule might represent an effort by the oocyst to mask itself with mosquito self-proteins to evade immune recognition[34,35]. A similar strategy is used by gametes that cover their surface with human factor H, which is present in the blood bolus, to evade human complement in the mosquito midgut[36]. In the present study, we found that oocysts could use their own protein, CSP, to avoid mosquito immune recognition. Previously, *Pfs47* on the ookinetes of mosquito-susceptible *P. falciparum* strain was demonstrated to block the JNK pathway to foster parasite development in the mosquito[11]. In contrast to *Pfs47*, concealing CSP is a strategy used by malaria parasite oocysts to evade host immunity as the mutation of the CSP pexel I/II domain of either *P. yoelii* or *P. berghei* resulted in the developmental arrest of mature oocysts (Fig. 1).

We found that the CSP pexel I/II domain mutation resulted in a conformational change (Fig. 5), which induced nitration of hemocytes attached to the midgut, and triggered mosquito immune responses. A similar phenomenon was observed when midgut epithelial cells were invaded by ookinetes, and it could be elicited by the exposure of epithelium intracellular molecules, which were sensed as danger signal by mosquito immunity[4]. Although the definitive mechanism by which the mutant parasite induces hemocyte nitration remains unknown, we postulated that the conformational change in CSP might change the physical surface of oocysts, which would be sensed by the circulating hemocytes because nitrated hemocytes were found to attach to mutant oocysts (Supplementary Fig. 8). After the hemocytes were activated, TEP1 released by hemocytes could more easily permeate into the mutant oocyst, as supported by our observation of the co-localization of TEP1 with CSP inside the oocyst (Fig. 2g). Although both NOX5 and HPX2 are essential for the nitration of ookinete-invaded midgut epithelium, only NOX5 was involved in the nitration reaction induced by mutant parasites (Fig. 4 and Supplementary Fig. 9), indicating that some differences might exist between these two mechanisms.

Previous research has demonstrated that CSP is required for sporozoite formation in oocysts through the disruption of the *CS* gene[37]. Another study showed that the lack of a repeated region, with or without the N-terminus, resulted in a defect in sporozoite development[38]. However, the deletion of the N-terminus (ΔNfull CSP mutant), or mutation of region II plus in the C-terminus of CSP, did not affect sporozoite formation in oocysts but severely

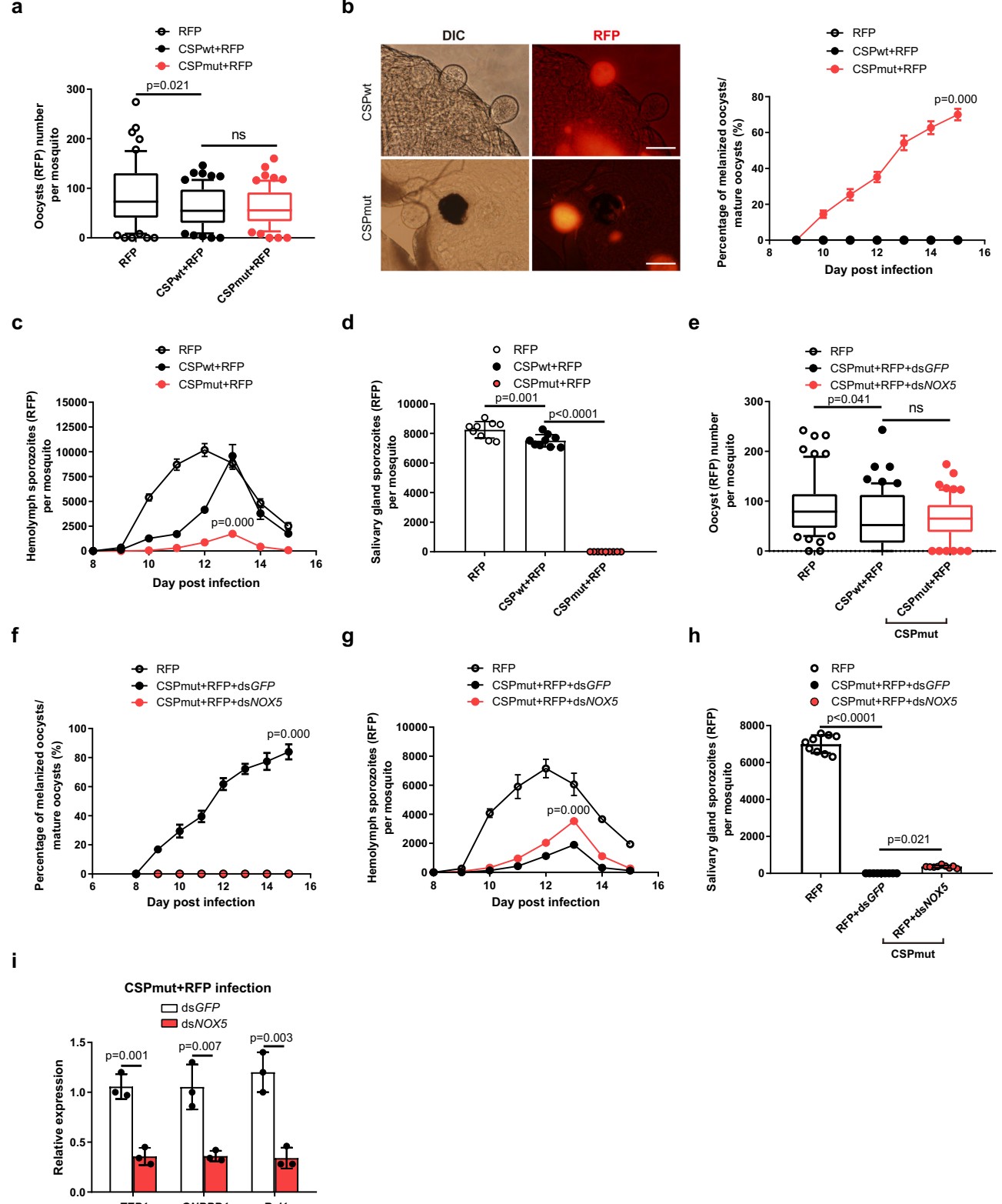

reduced the release of sporozoites from mutant oocysts[31,39]. In contrast, we revealed that CSP plays a role in the modulation of mosquito immune responses when its pexel I/II domain was mutated, which has not been reported in previous studies[31,38,39]. This discrepancy might be explained by the fact that different CSP domains were mutated in our study, and that CSP immune evasion may have been masked when the N-terminus or repeated region was deleted, or region II plus in the C-terminus was mutated in previous studies[31,38,39]. Alternatively, the melanization of mutant parasites might have been missed in previous studies[31,38,39]. Therefore, we described a dual function for CSP, including immune evasion and sporogonic development. Our finding is supported by a recently identified ookinete and sporozoite surface protein, *Plasmodium* Infection of the Mosquito

**Fig. 6 Pre-infection of mosquitoes with CSP$_{mut}$ reduces parasite burden when challenged with RFP-expressing WT parasites.** Mosquitoes infected with *Plasmodium yoelii* CSP$_{wt}$ or CSP$_{mut}$ parasites were re-challenged with *P. yoelii*-RFP 3 days after the first blood meal. **a** The average number of *P. yoelii*-RFP oocysts in mosquitoes ($n = 60$) pre-infected with or without *P. yoelii* CSP$_{wt}$ or CSP$_{mut}$ parasite at day 7 PI. **b** A representative image of *P. yoelii* oocysts in mosquitoes pre-infected with *P. yoelii* CSP$_{wt}$ or CSP$_{mut}$ parasites under the light microscope and using immunofluorescence, respectively (left); the percentage of melanized oocysts in mosquitoes ($n = 36$) pre-infected with *P. yoelii* CSP$_{wt}$ or CSP$_{mut}$ parasites at the indicated time points were compared (right); scale bar 50 μm. The average number of hemolymph sporozoites (**c**) in mosquitoes ($n = 45$), and salivary gland sporozoites (**d**) in mosquitoes ($n = 90$) pre-infected with *P. yoelii* CSP$_{wt}$ or CSP$_{mut}$ parasites were compared at the indicated times. The average number of *P. yoelii*-RFP oocysts (**e**) in mosquitoes ($n = 60$), the percentage of melanized oocysts (**f**) in mosquitoes ($n = 36$), the average number of hemolymph sporozoites (**g**) in mosquitoes ($n = 45$), and salivary gland sporozoites (**h**) in mosquitoes ($n = 60$) pre-infected with CSP$_{mut}$ parasite were compared after *NOX5* knockdown. **i** The mRNA levels of *TEP1*, *GNBP-B1*, and *Rel1* in mosquitoes ($n = 15$) pre-infected with *P. yoelii* CSP$_{mut}$ parasites at day 5 PI after *NOX5* knockdown were determined using real-time PCR. For box-plot diagram (**a**, **e**): middle line represents median; boxes extend from the 25th to 75th percentiles. The whiskers mark the 10th and 90th percentiles. A two-sided Student's *t*-test was used for the statistical analysis. The data are presented as the means ± SD; ns, no significance; *p*-values are shown. The data were combined based on three independent experiments (**a–i**). Source data are provided as a Source Data file.

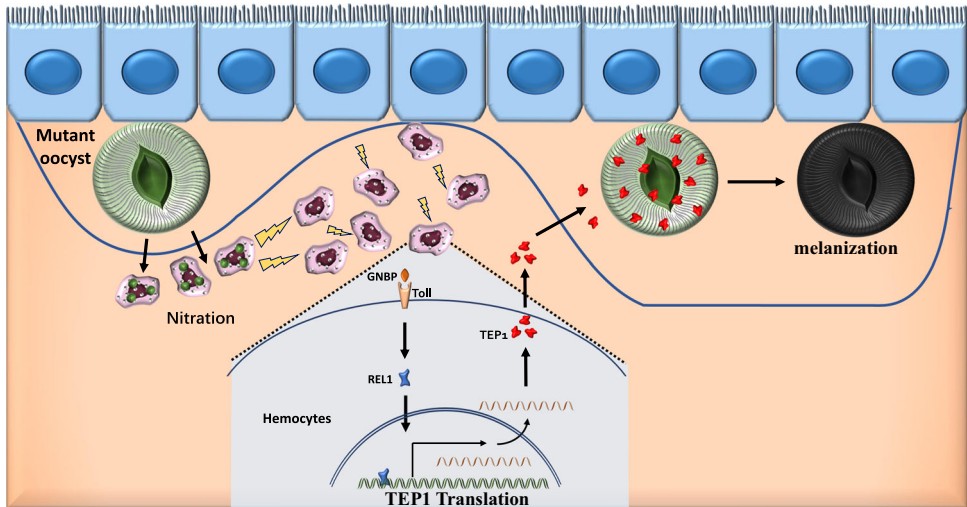

**Fig. 7 Schematic illustration showing how CSP$_{mut}$ parasites trigger mosquito immune responses.** CSP$_{mut}$ oocysts, with a conformational change in the inner membrane protein CSP due to a mutation in its pexel I/II domain, induce the nitration of hemocytes attached to the midgut and enhance the expression of TEP1 in a Toll pathway-dependent manner. The binding of TEP1 to oocysts results in the melanization or vacuolation of mature oocysts, and the release of sporozoites from the CSP$_{mut}$ oocysts was impaired.

Midgut Screen 43 (PIMMS43), which is also required for parasite evasion of the mosquito complement-like response, as well as sporogonic development in the oocyst[40].

We also noted that the number of sporozoites in the hemolymph from CSP$_{mut}$ parasite oocysts did not return to CSP$_{wt}$ parasite levels, even when the melanization of mutant oocysts was completely abrogated. As CSP plays a critical biological role in parasite development in mosquitoes, the effects of CSP pexel I/II domain mutation on sporozoite release from oocysts could not be completely excluded. Further, in addition to the conformational change in CSP in the inner membrane of the oocyst, the conformation of CSP on hemolymph sporozoites also varied (Fig. 5), which might induce a more robust mosquito immune response than that of CSP$_{wt}$ sporozoites. As a result, more mutant sporozoites might be cleared from the hemolymph than CSP$_{wt}$ sporozoites. However, this needs to be confirmed in future studies.

In conclusion, we found that mutation of the CSP pexel I/II domain induced the nitration of hemocytes, which subsequently recruited additional hemocytes to the midgut basal lamina. Furthermore, nitration enhanced the expression of TEP1 in hemocytes in a Toll-dependent manner, induced melanization of mature oocysts, and resulted in the impaired release of sporozoites from the CSP$_{mut}$ oocysts (Fig. 7). Thus, we described a TEP1-dependent immune response elicited by the recognition of

mutant mature oocysts. In contrast, previous studies have reported a significant loss in parasite numbers between the early and mature oocyst stages mediated by mosquito hemocytes[8,14,15] and PPOs[14,15] but independent of TEP1 function[13,41]. Therefore, our findings might represent an additional mechanism by which oocysts can be recognized and targeted in the mosquito. However, it is possible that the differences in the immune response between ours and previous studies were due to the different strains of *Anopheles* used. CSP is regarded as the predominant protective antigen of the irradiation-attenuated sporozoites[42], and the main component of the leading liver-stage subunit vaccine RTS,S[43,44]. Our results highlight CSP also as a potential target for malaria transmission-blocking vaccines.

## Methods

**Malaria parasites, mosquitoes and mice**. Both *P. yoelii* 265BY and *P. yoelii* 265BY-RFP, which was constructed using a single cross-mediated insertion of RFP in the *SSU rRNA* gene of the *P. yoelii* 265BY parasite[45], were maintained by alternate passaging between Kunming mice (Army Medical University, Chongqing, China) and *Anopheles stephensi* in our laboratory. *P. berghei* ANKA was donated by Dr. Xin-Zhuan Su (NIAID, MD, USA). *A. stephensi* from our laboratory was maintained at 27 °C at a relative humidity of 70–80%, and fed with a 10% sugar solution containing 0.5% para-aminobenzoic acid (PABA). The room was on a cycle of 12 h light/dark (lights off at 18:00).Kunming mice (Female, 6–8 weeks old) were provided by the Laboratory Animal Center of the Army medical University (Chongqing, China); BALB/c and C57BL/6 J mice (Female, 6–8 weeks old) purchased from Beijing Animal Institute (Beijing, China). The mice were kept in a

temperature- and humidity-controlled room on a cycle of 12 h light/dark (lights off at 19:00). All animal protocols were reviewed and approved by the Animal Ethics Committee of the Army Medical University Institute of Medical Research (AMUWEC20181777).

**Construction of the CSP mutant parasites**. Using CRISPR-Cas9 technology, *P. yoelii* 265BY parasites were generated by replacing the *P. yoelii* 265BY WT *CSP* sequence with a mutated *CSP* pexel I/II domain sequence (the sequences of pexel I and pexel II were mutated into ANANA and ALAGA, respectively), as previously reported[18]. Briefly, a single guide RNA (5'-GATTCTCTACTTCCAGGATA-3') specifically targeting the *P. yoelii* 265BY *CSP* gene (GenBank Gene ID: DQ012939.1) was inserted downstream of the *Plasmodium* U6 promoter in the pYC plasmid (donated by Dr. Jing Yuan, Xia'men University, China). The homologous recombinant fragment, containing a 5' untranslated region (551 bp) and a fragment encoding the mutated *CSP* pexel I/II domain (1284 bp), was constructed by overlapping PCR and then inserted into the multiple cloning sites of the pYC plasmid (Supplementary Fig. 1). The resulting pYC-CSP$_{mut}$ recombinant plasmid was amplified, purified using the E.Z.N.A® Endo-Free Plasmid Midi Kit (Omega Bio-Tek, GA, USA), and electroporated into synchronic schizont cultures as previously reported[46]. Following electroporation, the transfection medium was immediately injected into the tail vein of Kunming mice.

Once parasites appeared in the blood, mice were fed pyrimethamine (6 μg/mL; Sigma-Aldrich, St. Louis, MO, USA). The pyrimethamine-resistant parasites were harvested, and then each mouse was injected intravenously with 100 μL of parasite solution with ~1.0 infected red blood cell in phosphate-buffered saline (PBS) to clone pyrimethamine-resistant parasites. After 7–9 days, blood from mice with cloned parasites was harvested, and genomic DNA was extracted. The mutated CSP$_{mut}$ was identified using PCR and then verified using DNA sequencing (Supplementary Fig. 1). The construction of the *CSP* mutant in *P. berghei* ANKA parasites was performed as previously described[19].

**Infection of mice with malaria parasites**. Female 6–8-week-old BALB/c mice (5 mice each group) were intravenously infected with $1 \times 10^6$ *P. yoelii* 265BY CSP$_{wt}$ or CSP$_{mut}$ parasitized red blood cells (pRBCs) and *P. berghei* ANKA CSP$_{wt}$ or CSP$_{mut}$ pRBCs, and the parasitemia and gametocytemia were recorded daily.

**Mosquito infection**. For infection with CSP$_{wt}$ or CSP$_{mut}$, 3–5-day-old female *A. stephensi* mosquitoes were allowed to feed on BALB/c mice (3–5 mice each group) displaying over 0.1% gametocytemia, and then fed with a 10 % sugar solution at 23–24 °C. On day 7, mosquito midguts were dissected to determine the oocyst burden. Hemolymph was harvested on days 8–15 to determine the number of hemolymph sporozoites, and salivary glands were dissected at day 17 post blood meal.

**Quantitative real-time PCR analysis**. Total RNA was extracted from either CSP$_{wt}$- or CSP$_{mut}$-infected mosquitoes using TRIzol$^{TM}$ Reagent (Thermo Fisher Scientific, Waltham, MA, USA) following the manufacturer's instructions. The quality and concentration of each RNA sample were determined using a NanoDrop One Spectrophotometer (Thermo Fisher Scientific), and 2 μg RNA was used for first-strand cDNA synthesis with the PrimeScript™ RT reagent Kit with gDNA Eraser (Perfect Real Time) (Takara Bio Inc., Japan). The resulting cDNA was then used as a template for subsequent quantitative real-time PCR (qPCR) using the TB Green® Premix Ex Taq™II (Tli RNaseH Plus) (Takara Bio Inc.) with target-specific primers (Supplementary Table 3). The assay was run on a CFX96 Real-Time PCR Detection System (Bio-Rad, Hercules, CA, USA), and relative quantitative results were normalized to the ribosomal protein S7 gene which was used as an internal control.

**dsRNA synthesis and gene silencing**. All the dsRNA products for silencing genes such as *TEP1*, *Rel1*, *GNBP-B1*, and *NOX5* were generated using a cDNA template from *A. stephensi*, and the MEGAscript™ RNAi Kit (Thermo Fisher Scientific) as previously described[3,7]. In brief, 3-day-old female *A. stephensi* were cold-anesthetized and injected with 3 mg/mL of dsRNA using a Nanoject II injector (Drummond Scientific Co., Broomall, PA, USA), 2 days prior to infection. Silencing efficiency was examined at day 2 post-knockdown by qPCR, as described above. Primers designed for specific transcript targets are listed in Supplementary Table 3.

**Electron microscopy**. Midguts dissected from infected mosquitoes at day 11 after blood meal were fixed with 2.5% (v/v) glutaraldehyde in 0.05 M PBS (pH 7.4) with 4% sucrose for 12 h, and post-fixed in 1% osmium tetroxide for 1 h. After 30 min of en bloc staining with 1% aqueous uranyl acetate, the midgut tissues were dehydrated in ascending concentrations of ethanol and embedded in Epon 812. Ultrathin sections were stained with 2% uranyl acetate in 50% methanol and lead citrate and then observed under an electron microscope (Zeiss CEM 902; Zeiss Group, Oberkochen, Germany).

**Polyclonal antibody preparation**. Anti-CSP-N, anti-CSP-C, anti-CSP-repeat-region, and anti-TEP1 were synthesized by GeneCreate Biotech (Wuhan, China). Polypeptide and antigenic sequences designed for specific antibodies are listed in Supplementary Table 4. Peptides were purified by high-performance liquid chromatography (HPLC) on a Waters Corporation Prominence HPLC system (Milford, MA, USA), using a Waters X Bridge C18 column (4.6 × 250 mm × 5 μm; Waters Corporation), and then conjugated to bovine serum albumin (BSA). The conjugated peptides were emulsified with Freund's adjuvant (Sigma-Aldrich) and used to immunize specific-pathogen-free rabbits at 0, 2, 4, and 6 weeks. Seven days after the last immunization, serum was collected, and the antibody titers were determined using enzyme-linked immunosorbent assay (cat. no. 44-2404-21, Nunc 584 MaxiSorp flat bottom, Nalge Nunc International, Penfield, NY, USA). Protein G Sepharose (GE Healthcare, Chicago, IL, USA) was used to purify the polyclonal antibodies. For anti-*P. yoelii* HSP70 and anti-mosquito S7 polyclonal antibody preparation, the coding sequences of *P. yoelii* HSP70 (GenBank No. PY17X_0712100) and *A. stephensi* S7 (VectorBase Gene ID: ASTE004816) were separately cloned into pCzn1 (Zoonbio, Nanjing, China), and pET B2M (GeneCreate Biotech, Wuhan, China) expression vector, and the expression of both HSP70 and S7 in *E. coli* Arctic-ExpressTM was induced by IPTG, and the protein was purified by Ni column. New Zealand white rabbits were immunized with Hsp70 protein or S7 protein as antigen. After 3-4 rounds of immunization, the specific polyclonal antibody against HSP70 or S7 was purified by antigen affinity purification column, and the titer, purity, and concentration of the purified antibody were detected by ELISA. Experiments were carried out by Zoonbio and GeneCreate Biotech.

**Immunofluorescence analysis**. Hemocytes were harvested from mosquitoes infected with either CSP$_{wt}$ or CSP$_{mut}$ on day 5 post blood meal and used for the detection of TEP1 expression. To detect the change in CSP conformation, oocysts and hemolymph sporozoites were collected from mosquitoes infected with CSP$_{wt}$ or CSP$_{mut}$ on days 7 and 12 post blood meal, respectively. The hemocytes and sporozoites were then mounted on glass slides treated with poly-L-lysine and fixed with 4% paraformaldehyde for 1 h. The midguts for oocyst staining were directly transferred into EP tubes for fixation. All samples were washed, blocked with PBS/BSA, and incubated with primary antibodies diluted (1:200) in PBS at 4 °C overnight. Samples were washed 2–3 times with PBS and incubated for 2 h at room temperature with Cy3-labeled or Alexa 488-conjugated secondary antibodies (Beyotime Biotech, Nantong, China) diluted (1:500) in PBS. The following primary antibodies were used: anti-CSP-N, anti-CSP-C, anti-CSP-repeat region, and anti-TEP1. Cell nuclei were counterstained with 4,6-diamidino-2-Phenylindole (DAPI; Beyotime Biotech). Microscope slides were mounted using Dako Fluorescence Mounting Medium (Agilent).

**Observation of hemocytes attached to the midgut basal lamina**. To investigate the recruitment of hemocytes to the basal laminal of the midgut of mosquitoes infected with the CSP$_{mut}$, 3–5-day-old *A. stephensi* were injected with 69 nL of 100 μM Vybrant® CM-DiI cell labeling solution (Thermo Fisher Scientific), the day prior to gene-silencing experiments, as previously described[5]. To maintain the attachment of hemocytes to the midgut, 276 nL of 16% paraformaldehyde was injected into the anesthetized mosquitoes. The mosquitoes were allowed to stand for 40 s, and then the midgut was dissected in a 4% paraformaldehyde solution, and further fixed with 4% paraformaldehyde overnight at 4 °C. The following day, the midguts were washed twice with PBS, blocked with PBS containing 1% BSA for 40 min, and washed twice with the same solution. For actin and nuclei staining, midguts were incubated for 30 min at room temperature with 1 U of Alexa Fluor® 647 phalloidin (Thermo Fisher Scientific), and DAPI (Beyotime Biotech). Tissues were mounted on microscope slides using Dako Fluorescence Mounting Medium (Agilent). Hemocytes were visualized by confocal microscopy (Leica TCS SP8) and LAS AF Lite was used for image acquisition and export, the number of hemocytes per midgut in each biological condition was analyzed.

**Nitration assay**. For immunofluorescence analysis of nitration in hemocytes attached to the midgut basal lamina, Vybrant® CM-DiI cell labeling solution were injected into mosquitoes, and the midgut was dissected and fixed as described above. The following day, midguts were washed twice with PBS, blocked with PBS containing 1% BSA for 40 min, and washed twice with the same solution. The midguts were incubated overnight at 4 °C with diluted mouse anti-nitrotyrosine primary antibody (Abcam, Burlingame, CA, USA) diluted in PBS (1:500). Samples were then washed 2–3 times with PBS and incubated for 1 h at room temperature with Alexa 647-conjugated secondary antibodies (Beyotime Biotech) diluted (1:200) in PBS. Then, midguts were incubated with anti-TEP1 antibody diluted in PBS (1:200) for 3 h at room temperature. Samples were then washed 2–3 times with PBS and incubated for 1 h at room temperature with Alexa 488-conjugated secondary antibodies (Beyotime Biotech) diluted (1:500) in PBS. For nuclei staining, midguts were incubated for 5 min at room temperature with DAPI (Beyotime Biotech). Tissues were mounted on microscope slides using Dako Fluorescence Mounting Medium (Dako).

**In vitro assay of oocyst sporozoite release**. Midguts were dissected from either CSP$_{wt}$- or CSP$_{mut}$-infected mosquitoes on day 9 post blood meal. For each

experiment, 20 midguts were incubated at 23 °C in 200 μL of RPMI medium. The solution (10 μL) was removed every 30 min, and the sporozoites released in the supernatant were quantified with a hemocytometer.

**Western blot analysis**. For the detection of TEP1 activation, hemolymph was collected from mosquitoes ($n = 20$) infected with $CSP_{wt}$ or $CSP_{mut}$ on days 3, 5, and 7 after blood meal. For the detection of CSP expression, oocysts from $CSP_{wt}$ or $CSP_{mut}$ parasite-infected mosquito midguts ($n = 40$) were collected on days 4, 5, 7, 9, and 12 post blood meal. Samples were dissolved in T-PER Tissue Protein Extraction Reagent (Thermo Fisher Scientific) containing a Halt Protease & Phosphatase Inhibitor Cocktail (Thermo Fisher Scientific). Protein samples (~20 μg) were mixed with Laemmli Sample Buffer (Bio-Rad), heated at 100 °C for 10 min, and then separated on a 10% TGX Stain-Free FastCast™ Acrylamide Kit (Bio-Rad). The separated proteins were transferred to a PVDF membrane and blocked in BSA for 1 h at room temperature. For the detection of TEP1 activation, the membrane was then incubated with a 1:5000 dilution of rabbit anti-TEP1, and rabbit anti-S7 as an internal control, in TBST blocking buffer overnight at 4 °C; For the detection of CSP expression, the membrane was incubated with rabbit anti-CSP-repeat, with rabbit anti-plasmodium hsp70 as an internal control. Membranes were then washed three times for 5 min in TBST and incubated with secondary anti-rabbit horseradish peroxidase (1:5000, Zhongshan Golden Bridge Biotechnology, Beijing, China) for 1 h at room temperature. After washing in TBST, the membrane was visualized using Western BLoT Chemiluminescence HRP Substrate (Takara Biotech Inc.). For comparative analysis between samples, densitometric analysis was performed using the software ImageJ.

**In vitro activation of mosquito hemocytes by oocysts**. On day 5, midguts from either $CSP_{wt}$- or $CSP_{mut}$-infected mosquitoes ($n = 20$) with *NOX5* knockdown or not were dissected and homogenized. The homogenate and hemocytes from uninfected mosquitoes ($n = 20$) were incubated for 3 h at 28 °C, with medium containing antibiotics (including gentamicin, Sigma-Aldrich; penicillin/streptomycin/neomycin solution, Amphotericin B solution, Sangon, China). The medium containing antibiotics and uninfected midguts from mosquitoes were used as controls. After incubation, the incubation product was centrifuged at $500 \times g$ for 30 s to remove mosquito tissue, and then the co-cultured hemocytes were collected with a centrifuge at $2000 \times g$ for 5 min and smeared on coverslips pretreated with a poly-L-lysine solution. The immunofluorescence staining as described above was used to observe the nitration of hemocytes triggered by oocysts, and WGA (10 μg/mL, Invitrogen), which specifically marks mosquito hemocytes, was used to distinguish hemocytes from other mosquito cells. The mRNA levels of *GNBP-B1*, *Rel 1*, and *TEP1* in hemocytes collected from naïve or from ds*NOX5* mosquitoes were determined by quantitative real-time PCR analysis, as described above.

**Mosquito hemocyte depletion assay**. The day prior to $CSP_{mut}$ infection, naïve female mosquitoes (3–5 days old) were injected intrathoracically with either 69 nL of control liposomes or CLDs (Standard Macrophage Depletion Kit; Clodrosome® + Encapsome®; Encapsula Nano Sciences, Brentwood, TN, USA) using a Nanoject II injector (Drummond Scientific Co.). As previously described[22], 1:5 dilutions of liposomes and CLDs in 1 × PBS were used for the experiments. Following liposome injection, mosquitoes were infected with $CSP_{mut}$ parasites. The effect of clodronate liposomes on the depletion of mosquito hemocytes was determined by the expression of cell markers *NimB2* (hemocytes), *LRIM15* (granulocytes), and *SCRB9* (oenocytoids) with real-time PCR as previously reported[26]. The change in oocyst melanization, the number of sporozoites, and relative expression of *TEP1* were determined as described above.

**Statistical analysis**. All statistical analyses were performed using SPSS (IBM, Armonk, NY, USA). A Student's *t*-test was used for two groups if the data were normally distributed to compare continuous variables, and if not, then the Mann–Whitney *U* test was used for comparisons among groups. Pairwise differences in normally distributed variables were compared using the one-way analysis of variance (ANOVA) for multiple comparisons. Significance was set at $p < 0.05$. Error bars represent the standard deviation (SD).

**Reporting summary**. Further information on research design is available in the Nature Research Reporting Summary linked to this article.

## Data availability

Source data are provided with this paper. RNA-sequencing data has been deposited in NCBI's Gene Expression Omnibus and is accessible through GEO Series accession number GSE176061.

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

## Acknowledgements

This work was supported by the National Natural Science Foundation of China (No. 81772226, J.Z. & No. 81601783, F.Z.), the State Key Program of the National Natural Science Foundation of China (No. 81830067, W.X.), and the Natural Science Foundation of Chong Qing (No. cstc2018jcyjAX0609, J.Z.). We would also like to thank Editage (www.editage.cn) for English language editing.

## Author contributions

W.X., J.Z., and F.Z. designed the study. F.Z., J.Z., H.Z, S.L., K.Z., X.Q., Y.F., J.Z., T.L., L.W., and X.L. performed the experiments. H.Z., F.Z., and J.Z. analyzed the data. W.X., J.Z., and F.Z. wrote the manuscript.

## Competing interests

The authors declare no competing interests.
