## [Peer Review File · Nature Communications]

Malaria oocysts require circumsporozoite protein to evade mosquito immunityReviewers' Comments:

Reviewer #1:

Remarks to the Author:

Key results

This paper addresses the interaction of the mosquito with the malaria parasite oocyst. The authors generated a mutant of a rodent malaria parasite which expresses the circumsporozoite protein (CSP) with point mutations in two motifs (pexel I and pexel II). CSP has been well studied and is known to play an essential role in the oocyst and sporozoite stages. This mutant has a novel phenotype, in that while oocysts develop they are melanized by the mosquito immune system. This led the authors to hypothesize that in wild type parasites CSP acts in protecting the oocysts from attacks of the immune system. They then investigate in great detail the mosquito factors that lead to the melanization of the mutant oocysts.

Validity

The experiments are carried out with proper controls and repeated at least twice. The statistical analyses show that the data are robust and conclusions are valid.

Significance

I find this to be an interesting and significant paper. The study reveals a new role of CSP in protecting the parasite from the immune system. It also sheds new light on how the mosquito immune system recognizes and attacks the malaria parasite. The paper adds important new information on the interaction of parasite and mosquito in the mosquito midgut stage. The study is very well carried out, and the experiments are appropriate to address the scientific questions arising.

Data and methodology

The paper uses different approaches to investigate the interaction of oocyst/mosquito immunity. The data is clearly presented, the necessary controls are included, and the methods well described. All methods are described in detail.

Analytical approach

The statistical analysis is appropriate and described in each case.

Suggested improvements

One issue that I found puzzling is that a similar mutant with point mutations in the two pexel motifs has been previously described (Singh et al, 2007). In that case sporozoites developed normally in the mosquito. In this paper, the pexel motifs are not described in any detail and the reason why these specific point mutations were chosen is not discussed. It is important that the authors comment on the differences between their new mutant and that described by Singh et al, especially as relate to the phenotype. In this study in each pexel motif there are three point mutations while in Singh study there were two, how would this influence the CSP conformation? For clarity I suggest that in Fig. 1 the pexel motifs are shown as amino acid sequences with indications which amino acids were mutated. The authors mention that their mutated line can be used for transmission blocking. I would like to see this expanded, as the reasoning is not explained.

Previous work (mostly by Barillas-Mury) has suggested that the epithelial surface is nitrated in response to parasite invasion, here it is stated that the hemocytes themselves are nitrated these differences in observation should be discussed.

Line 148-150 states that some oocysts developed normally and produced thousands of sporozoites, but in the figures the number of mutant sporozoites are depicted as zero at all time points, which however is due to mislabeling of the Y-axis. For example, the scale in the sporozoite graphs in Fig. 1 f-i is not correct. The lowest point on the Y-axis is not zero as the other points are integers of 105. Same in 2d, 6c, 6g, 6h. In figures 3d and 4a on top of the heatmaps tendency and pathways are mislabeled. In 4a all genes shown are upregulated. The genes in the heatmaps should be listed with their gene id's and their gene names in Supplementary Material.

Clarity and context

The paper is generally well written, and experiments and conclusions are mostly clearly described. Previous work is reported to put this study in context, except that there is little discussion on the pexel motifs (as I discuss above).

References

Reference #1 is not the correct one in this context, and should be replaced by Kumar et al, 2010 (reference #25). References #14 and #23 does not contain the method referred to. I suggest the authors check that all references are correct.

Your expertise

The paper was within my area of expertise.

Reviewer #2:

Remarks to the Author:

Review of Zhu et al.

This is a manuscript of significant interest to both the parasite and vector community, highlighting new roles of CSP for immune evasion in the mosquito vector. However, my enthusiasm is significantly dampened by the lack of experimental details, rigor, and biological context provided in the study. There are also conflicting results within the data presented and issues with the experimental design that do not fully enable the conclusions proposed by the authors. As a result, significant work needs to be performed before this manuscript can be considered for publication.

Major comments

-An enhanced discussion of described roles of CSP and its phenotypes on oocyst development (localization in the inner membrane complex, SPZ production, SPZ egress) would greatly benefit the manuscript and better place the results in the greater context of experiments that have been previously performed in the mosquito host and the oocyst phenotypes they produced. Although oocyst melanization was not previously reported, I think it is also important to mention the ΔN_{full} CSP mutant (Coppi et al 2011-JEM) which should encapsulate the PEXEL mutations in the proposed work.

-Throughout the manuscript, the manner in which the data are present in the results text and figure legends lack experimental detail, such that at present the reader needs to hunt down the methods and resulting outcomes to fully interpret the presented information. In other cases, this is not possible. For example, if we examine Fig 1e it is unclear what "n" means. I assume that this is 6 individual mosquitoes examined at each time point, but how many oocysts were used to collect data. This is obviously a variable number. This information was not found in the methods. Similarly, for Fig1f-I, does an n=30 mean that 30 mosquitoes were examined for each individual timepoint? For Fig 2g, there is no mention of what day the IFA was imaged. This is critical information in the evaluation of the work. While taken individually, these are minor details, but since these are present to some degree throughout the manuscript, this lack of experimental detail makes interpretation more difficult and lessens confidence in the rigor of the work. Furthermore, several figure panels have only n=12, or infections with n=20, which is not very robust and questions the number of independent replicates that were performed.

-In Figure 1, there is a complete absence of hemolymph/salivary sporozoites in infections with the CSPmut in both parasite species. Yet, in subsequent figures, the CSPmut produces reduced, yet measurable sporozoite numbers. There is no mention of this and seems to erode at the foundation of this work.

-Why are mosquitoes maintained on PABA in the lab colony? This may artificially promote melanization by increasing the availability of quinone or ROS substrates used in melanin biosynthesis.

-A major issue with the current manuscript is the omission of several key points of biology that have seemingly been ignored that would greatly improve upon the context of the findings:

1. Expression/roles of CSP. Based on the ability of the mosquito immune system to recognize the CSPmut and the role of the PEXEL motifs that were modified, it suggests that the role of the CSPmut would be acting on establishing the integrity of the inner membrane complex of the parasite and this

potentially allowing recognition of the oocyst at later stages of development. However, I am confounded by the presence of CSPmut phenotypes at Day 5 or earlier in the manuscript (such as with TEP1), when previous studies have described CSP expression at Day 6/7 in Py (Aikawa et al 1990). As a result, I think it is important to more closely demonstrate the exact expression of CSP in Py in these experiments. I find it difficult to believe that a phenotype can be produced without actual expression of the protein.

2. PPOs and oocyst killing. Several lines of evidence have implicated hemocyte derived PPOs in the mechanisms that mediate oocyst killing (Kwon and Smith, 2019-PNAS; Kwon et al 2020-bioRxiv), yet these are not discussed. This seems highly relevant since PPOs are precursors that are believed to initiate melanin biosynthesis, an integral component of the CSPmut phenotype presented by the authors. In addition, some thought should be placed into how the much earlier non-melanizing oocyst killing roles could later translate to melanization in the CSPmut background.

-While parasite infections can be highly variable, I have concerns regarding the numbers of hemolymph sporozoites presented in the manuscript, which are in excess of 60,000 spz in many experiments, while in others range from 5-15,000. The latter number is much more in line with previously published studies.

-The data presented in Figure 3 are not very robust. Based on morphology, the oenocytoids presented in Fig3b are not oenocytoids. Oenocytoids are rounded, without any extensions. This confounds the data presented in Figure 3b, where in addition both oenocytoid and prohemocyte are misspelled in the figure. I further doubt the data presented since the combined % of granulocytes and oenocytoids in Fig3b exceed 100%. In addition, >90% of cells following fixation should be granulocytes.

-For the data presented in Figure 3D and Figure 4A, there is little description of the heatmaps presented. My interpretation is that only two replicates ("groups") are presented for each biological condition, and there is a great deal of variance between them. Furthermore, it would be beneficial to include gene abbreviations where possible for the individual genes included in the analysis to interpret the data without having to input every accession #. At present these figures provide little value.

-When paired with the issues of hemocyte descriptions in Figure 3b, the data presented in Fig4C are not very convincing that hemocytes undergo nitration. The DiI-labeling does not overlap very well in the images shown, and am not convinced that these are indeed hemocytes. Additional work is required to more accurately demonstrate that a significant portion of hemocytes undergo nitration by quantifying co-localization experiments with DiI and NY. Additionally, it may be interesting to follow-up with TEP1 and NY co-localization experiments as a reflection of immune activation. Are these immune activated or are they dead/dying in response to infection? It also would be interesting to know if there are potential differences in nitration between circulating and sessile (attached to midgut) hemocytes.

-For the data presented in Figure 4C, do these reported hemocytes bind to the midgut in close proximity to oocysts? This seems relevant for their potential role in oocyst melanization and should be examined.

-The in vitro hemocyte activation studies presented in Figure 5 are very difficult to interpret given the lack of experimental details provided. There is mention that midguts were treated with antibiotics, how was this done? How were the hemocytes collected? How many mosquitoes were used for each experiment to obtain hemocytes? If hemocytes were incubated with midgut homogenates, how were hemocytes identified? If I understand what was done, I would argue that these experiments are very artificial and may not accurately identify hemocytes from the rest of the homogenate. Furthermore, the article for which the methodology is derived from (Levashina et al 2001) does not mention a similar assay (they only use mosquito hemocyte-like cell lines) and wonder if this is the correct reference.

-I have several comments regarding the double infection experiments in Figure 6. First, evidence suggests that priming with a parasite reduces subsequent infections (Rodrigues et al 2010-Science), thus supporting the decrease RFP parasite infection intensity in those previously challenged mosquitoes. However, unlike Rodrigues et al. the parasites were allowed to persist confounding the results. At the time of re-challenge, the initial infection (3 days post-infection) would not have any influence since CSP would not be expressed. As a result, any effects on RFP parasite melanization would be a "by-product" of persisting CSPmut parasites. Furthermore, the numbers of non-RFP parasites (CSPwt or CSPmut) should also be counted. Moreover, as oocysts are killed by melanization or other means, they lose their fluorescence, such that RFP parasites can no longer be distinguished from melanized CSPmut parasites. Therefore, a clear determination of the origin of the melanized parasites cannot be determined from the experiments displayed for Figure 6.

Minor comments

-Use of arrested to describe CSP mut throughout manuscript is somewhat misleading. Oocysts are formed, such that their development is not arrested, they develop normally. The phenotypes presented impair the ability of spz to ultimately reach the salivary gland. In my mind, this is an important distinction and can be misinterpreted. This terminology should be changed in the revised manuscript (line 101, 123, 156,277, 406,444,458)

-Since mutations of both Py and PB are described in Figure 1, there is some confusion as to which species the remaining experiments are performed with. This should be transparent to the reader throughout the manuscript and would at least restate this in the figure legends.

-I assume that infected mosquitoes are maintained on 10% sucrose after infection, but this is not directly stated. Please address.

-The display of hemolymph spz is inconsistent through out the manuscript, where numbers are displayed as scientific numbers or whole numbers.

-For Figure 2C there is no mention in the figure legend what the white arrows are pointing to. The morphology of the developing oocysts seem different, but this really isn't expanded on. Furthermore, are we to assume that if an oocyst is melanized (Fig2a), then the internal sporozoites (Fig2c) are melanized as well?

-Was there any validation that the clodronate liposome treatment depletes phagocytic granulocytes in the proposed experiments?

-For Figure 4d, adding example images of nitration differences in cells across different conditions would add to the rigor of these results.

-For gene expression analysis throughout the manuscript, are these whole mosquitoes? Or specific tissues? Moreover, it also isn't always transparent what day of infection this analysis is being performed. This information should be clearly specified.

-There is mention of vacuolation of oocysts in Figure 7, yet only briefly mentioned for Fig2 but this isn't displayed in the manuscript. This should either be amended or removed.

-In the methods for the nitration assays, there is mention of using the nitrotyrosine antibody at a 1:100 dilution or a 1:3000 dilution. These are wildly different and if an antibody react at a 1:3000 dilution, using it at 1:100 will likely produce an artificial signal. Please address.

Specific comments

-Line 78: While granulocytes have been implicated in the production of HdMV, this has not been

conclusively proven. I suggest changing this phrasing to "hemocytes".

-Lines 82-84: Please modify to "PG stimulates the production of HDF". Aside from a model proposed by Barletta et al 2019, there is no evidence that the components of HDF are actually produced in the fat body.

-Lines 85-87: Please remove "a gene belonging to the susceptible Plasmodium falciparum strain". Mosquitoes are susceptible/resistant, not parasites.

-Figure 2d: The y-axis describes hemolymph spz, yet the text describes in vitro experiments. Please correct.

-Figure 2G: Y axis is labeled "oocytes" instead of oocysts.

-Lines 195-196: While hemocytes undoubtedly produce TEP1, Volohonsky et al (2017 -Plos Pathogens) argue that TEP1 is predominantly produced in the fat body.

-Lines 308-309: Delete "and induce oxidative stress". A conformational change would promote immune detection first, oxidative stress could be a downstream response.

-Lines 591 and 602: The name of the antibody is "nitrotyrosine". Please correct.

Reviewer #3:

Remarks to the Author:

In this report, the authors show that when the conformation of the major sporozoite surface protein CSP is altered, the mosquito immune system is activated, resulting in death of oocysts. The authors provide evidence that activation of the immune system involves hemocyte nitration and upregulation of several genes in the Toll pathway, including the complement-like TEP1 gene.

- A major recommendation relates to rigor of data reporting. The authors state that several independent experiments have been conducted but present the results of only one "representative" (presumably the best one) experiment. I strongly feel that all data should be reported. That is, in each panel, the POOLED data of all the biological repeats should be reported, stating the actual number of biological repeats and the total number of entities (mosquitoes, guts, sporozoites, oocysts, etc.) that were assayed. This does NOT involve doing new experiments, but simply re-analyzing existing data.
- No strong evidence is presented that the mutant parasite could be used for transmission-blocking vaccine and statements to this effect in the abstract, introduction and discussion should be omitted.
- A comment is needed to explain how CSP with a defective pexel domain reaches the surface of the oocyst (Fig. 2g) and of the sporozoite (Fig. 5b).
- In some experiments, very large number of mutant sporozoites are found in the hemolymph (e.g., Fig. 3c, 4g). Issues that need to be addressed: 1) with what efficiency (if any) do these sporozoites invade the salivary glands? 2) are these sporozoites (either from salivary glands or from hemolymph) infective to mice? 3) is the motility of the mutant sporozoites affected? Data addressing these questions would significantly enhance the value of the findings.
- The Western blot of Fig. 2f uses 'S7' as "an internal control". This is puzzling, as S7 is a ribosomal protein. How can it be found in the hemolymph in large amounts and used as a reference? What is the source of anti-S7 antibody?

- An apparent inconsistency: in Fig. 1g no hemolymph sporozoites were detected, while in Fig. 2i several thousand sporozoites were detected.
- Nature Communications has a broad readership and articles should be easily accessible to the average reader. Figures 3d and 4a are not easy to understand. Perhaps a more extensive legend might help.
- Supplementary Table 1 needs a legend. What are the percentages? Are the values relative to dsGFP?
- Fig. 3b and Methods. What criteria were used to distinguish granulocytes from oenocytes?
- Fig. 3f and 3g. The legend says "... in WT or mutant parasite infected mosquitoes". Are the data shown for WT or mutant? One of the two appears to be missing.
- Legend of Fig. 4d. Explain how measurements were done.
- Fig. 5a and Methods. What is the origin of the anti-HSP70 antibody?
- Figs. 6a, 6d 6f and 6h: what is "KM"?
- Figs. 6b and 6e. I could not understand these figures. Are the oocyst numbers a sum of those originating from the first and the second infection? Can fluorescence be detected in melanized oocysts? Three curves are apparently plotted but only two are visible.
- Fig. 6d. After spending much time trying to understand, it appears that the label is wrong: middle black bar should be labeled CSPWT+RFP, not CSPmut+RFP, is this correct?
- Fig. 6e: Three curves are apparently plotted but only two are visible. Is dsNOX5 100% effective in suppressing melanization (cf. Fig. 5e)?
- Fig. 6e: it is confusing; the figure says "percentage of melanized oocysts/mature oocysts (%)" and the legend says "average number of P.y BY265-RFP oocysts".
- Fig. 6f: it is confusing; the figure says "oocyst number (RFP)" and the legend says "percentage of melanized oocysts"
- Fig. 7. A key component of the proposed model is that TEP1 is the effector that causes oocyst death. This assumption is easily testable but knockdown of TEP1 experiments have not been conducted. This is a significant weakness.
- Improvement of the English writing may be required.

In summary, the manuscript presents interesting experiments that report on a novel way that mosquito immunity is activated. However, weaknesses are noted and the lack of rigor in assembling the manuscript is disappointing, to say the least.

REVIEWER COMMENTS

Reviewer #1 (Remarks to the Author):

Key results

This paper addresses the interaction of the mosquito with the malaria parasite oocyst. The authors generated a mutant of a rodent malaria parasite which expresses the circumsporozoite protein (CSP) with point mutations in two motifs (pexel I and pexel II). CSP has been well studied and is known to play an essential role in the oocyst and sporozoite stages. This mutant has a novel phenotype, in that while oocysts develop they are melanized by the mosquito immune system. This led the authors to hypothesize that in wild type parasites CSP acts in protecting the oocysts from attacks of the immune system. They then investigate in great detail the mosquito factors that lead to the melanization of the mutant oocysts.

Validity

The experiments are carried out with proper controls and repeated at least twice. The statistical analyses show that the data are robust and conclusions are valid.

Significance

I find this to be an interesting and significant paper. The study reveals a new role of CSP in protecting the parasite from the immune system. It also sheds new light on how the mosquito immune system recognizes and attacks the malaria parasite. The paper adds important new information on the interaction of parasite and mosquito in the mosquito midgut stage. The study is very well carried out, and the experiments are appropriate to address the scientific questions arising.

Data and methodology

The paper uses different approaches to investigate the interaction of oocyst/mosquito immunity. The data is clearly presented, the necessary controls are included, and the methods well described. All methods are described in detail.

Analytical approach

The statistical analysis is appropriate and described in each case.

Suggested improvements

One issue that I found puzzling is that a similar mutant with point mutations in the two pexel motifs has been previously described (Singh et al, 2007). In that case sporozoites developed normally in the mosquito. In this paper, the pexel motifs are not described in any detail and the reason why these specific point mutations were chosen is not discussed. It is important that the authors comment on the differences between their new mutant and that described by Singh et al, especially as relate to the phenotype. In this study in each pexel motif there are three point mutations while in Singh study there were two, how would this influence the CSP conformation?

Response: Thanks. We originally aim to obtain CSP pexel mutant *P. yoelii* sporozoites for our liver-stage study. As the method to construct *P. berghei* CSP pexel mutant parasite in Singh et al, 2007 is not very clear for us, we consulted with the Dr. Singh. He told us that he is not sure how to mutate the CSP pexel domain of *P. yoelii*, because he did not mutate this strain before. He suggested us to mutate the first, third and fifth amino acids, as a SCIENCE paper in 2004 (MARTI et. al. 2004) demonstrated all those three amino acids are very important for pexel domain. However, as a result, we found no salivary gland sporozoites generated in mosquitoes infected with the mutant *P. yoelii*. Similar mutation *P. berghei* also result in the generation of few salivary

gland sporozoites. Therefore, we now realize that it is better to cite the paper of Science (MARTI et. al. 2004). Now, the cited reference was corrected, and we have described the mutated amino acids in details in both text and method.

According to the mutation approach in Singh et al, 2007, two points mutation in pexel I and II domain did not result in sporozoites development defect. We postulated this kind of mutation would not influence CSP conformation.

For clarity I suggest that in Fig. 1 the pexel motifs are shown as amino acid sequences with indications which amino acids were mutated.

Response: Thanks. The mutated amino acids have been indicated in Fig.1 now.

The authors mention that their mutated line can be used for transmission blocking. I would like to see this expanded, as the reasoning is not explained.

Response: Thanks. We demonstrated that the infection of mutant parasite could resist a challenge of WT parasite, indicating its potential for malaria transmission blocking. However, it is impractical. Therefore, the statement of the potential role of mutant parasite for transmission-blocking vaccine in the abstract, introduction and discussion now has been deleted.

Previous work (mostly by Barillas-Mury) has suggested that the epithelial surface is nitrated in response to parasite invasion, here it is stated that the hemocytes themselves are nitrated these differences in observation should be discussed.

Response: Thanks. The difference between hemocytes nitration in our study and epithelial cell nitration in previous work has been discussed.

Line 148-150 states that some oocysts developed normally and produced thousands of sporozoites, but in the figures the number of mutant sporozoites are depicted as zero at all time points, which however is due to mislabeling of the Y-axis. For example, the scale in the sporozoite graphs in Fig. 1 f-i is not correct. The lowest point on the Y-axis is not zero as the other points are integers of 105. Same in 2d, 6c, 6g, 6h.

Response: Thanks. You are right. Actually, both hemolymph and salivary gland sporozoites are developed in the mosquito, but the number is very low, as compared to the WT sporozoites. Fig 1 has been modified to show the low number of both hemolymph and salivary gland sporozoite.

In figures 3d and 4a on top of the heatmaps tendency and pathways are mislabeled. In 4a all genes shown are upregulated. The genes in the heatmaps should be listed with their gene id's and their gene names in Supplementary Material.

Response: Thanks, it has been corrected. Both Fig. 3d and 4a have been replaced by new ones, the genes of which have been indicated with corresponding gene ID and the names of genes.

Clarity and context

The paper is generally well written, and experiments and conclusions are mostly clearly described. Previous work is reported to put this study in context, except that there is little discussion on the pexel motifs (as I discuss above).

Response: Thanks for reviewer's appreciation, and the discussion of pexel motifs has been added.

References

Reference #1 is not the correct one in this context, and should be replaced by Kumar et al, 2010 (reference #25). References #14 and #23 does not contain the method referred to. I suggest the authors check that all references are correct.

Response: Thanks. We have carefully checked the references, and corrected wrong ones.

Reviewer #2 (Remarks to the Author):

Review of Zhu et al.

This is a manuscript of significant interest to both the parasite and vector community, highlighting new roles of CSP for immune evasion in the mosquito vector. However, my enthusiasm is significantly dampened by the lack of experimental details, rigor, and biological context provided in the study. There are also conflicting results within the data presented and issues with the experimental design that do not fully enable the conclusions proposed by the authors. As a result, significant work needs to be performed before this manuscript can be considered for publication.

Major comments

-An enhanced discussion of described roles of CSP and its phenotypes on oocyst development (localization in the inner membrane complex, SPZ production, SPZ egress) would greatly benefit the manuscript and better place the results in the greater context of experiments that have been previously performed in the mosquito host and the oocyst phenotypes they produced. Although oocyst melanization was not previously reported, I think it is also important to mention the Δ Nfull CSP mutant (Coppi et al 2011-JEM) which should encapsulate the PEXEL mutations in the proposed work.

Response: Thanks. We are not sure whether the oocyst of Δ Nfull CSP mutant was theoretically melanized or not, but it has been discussed, as well as the relation between different mutation of CSP and its phenotypes on sporozoite production and egress. Please refer to the revised manuscript.

-Throughout the manuscript, the manner in which the data are present in the results text and figure legends lack experimental detail, such that at present the reader needs to hunt down the methods and resulting outcomes to fully interpret the presented information. In other cases, this is not possible. For example, if we examine Fig 1e it is unclear what “n” means. I assume that this is 6 individual mosquitoes examined at each time point, but how many oocysts were used to collect data. This is obviously a variable number. This information was not found in the methods. Similarly, for Fig1f-l, does an n=30 mean that 30 mosquitoes were examined for each individual timepoint? For Fig 2g, there is no mention of what day the IFA was imaged. This is critical information in the evaluation of the work. While taken individually, these are minor details, but since these are present to some degree throughout the manuscript, this lack of experimental detail makes interpretation more difficult and lessens confidence in the rigor of the work. Furthermore, several figure panels have only n=12, or infections with n=20, which is not very robust and questions the number of independent replicates that were performed.

Response: Thanks. We have checked all experimental detail in both text and methods and added necessary information accordingly for reader to understand. For example, what is the meaning of “n” has been explained. The day for the performing IFA in Fig 2g has also been added.

We did repeat our experiments. As dsRNA injection always results in a relative high percentage of dead mosquitoes, and several time points of hemolymph and salivary gland sporozoites need to be counted, the number of mosquitoes used to count hemolymph or salivary gland sporozoites was not enough in some Figures. According to the suggestion of reviewer 3, we, therefore, have pooled three repeated data in most Figures now.

-In Figure 1, there is a complete absence of hemolymph/salivary sporozoites in infections with the CSPmut in both parasite species. Yet, in subsequent figures, the CSPmut produces reduced, yet measurable sporozoite numbers. There is no mention of this and seems to erode at the foundation of this work.

Response: Thanks. Actually, both *P.yoelii* and *P.berghei* mutant hemolymph sporozoites can be detected but the number is too low to be seen, as compared to the WT sporozoites. Although a few *P.berghei* mutant sporozoites are generated in salivary gland, but no mutant *Pyoelii* sporozoites are detected in the salivary gland. Fig 1 has been modified to show the low number of both hemolymph and salivary gland sporozoite now.

-Why are mosquitoes maintained on PABA in the lab colony? This may artificially promote melanization by increasing the availability of quinone or ROS substrates used in melanin biosynthesis.

Response: Thanks for your reminding. The supplement of PABA was originally to promote the synthesis of DNA for malaria parasite development in mosquitoes. We do not know PABA will increase the availability of quinone or ROS substrates for melanin biosynthesis. Therefore, we investigated the effect of PABA on the melanization of oocyst in mosquitoes infected with CSP_{mut} parasite, and no significant difference was found in the percentage of melanized oocyst in CSPmut-infected mosquitoes maintained on PABA or not, which is indicated as the following table (Table R1).

Table R1. The effect of PABA on the melanization of oocyst in mosquitoes infected with CSP_{mut} parasite

	D9	D12	D15
CSPwt	0%	0%	0%
CSPwt+PABA	0%	0%	0%
CSPmut	26.09%(±7.8)	47.91%(±8.8)	62.7%(±17.01)
CSPmut+PABA	24.53%(±7.02)	50.59%(±7.3)	66.59%(±16.6)

-A major issue with the current manuscript is the omission of several key points of biology that have seemingly been ignored that would greatly improve upon the context of the findings:

1. Expression/roles of CSP. Based on the ability of the mosquito immune system to recognize the CSPmut and the role of the PEXEL motifs that were modified, it suggests that the role of the CSPmut would be acting on establishing the integrity of the inner membrane complex of the parasite and this potentially allowing recognition of the oocyst at later stages of development.

However, I am confounded by the presence of CSPmut phenotypes at Day 5 or earlier in the manuscript (such as with TEP1), when previous studies have described CSP expression at Day 6/7 in Py (Aikawa et al 1990). As a result, I think it is important to more closely demonstrate the exact expression of CSP in Py in these experiments. I find it difficult to believe that a phenotype can be produced without actual expression of the protein.

Response: Thanks a lot for the suggestion. We have investigated the CSP expression in mosquitoes at different time points after infected with *Pyoelii*, and found that the expression of CSP (~70KDa) in the infected mosquitoes was could be detected by western blot at d5 PI, and it was detected as early as 4-5 days after infection by real-time PCR. This is earlier than what is previously reported (Aikawa et al 1990). The discrepancy might be explained by different method in ours and in previous. In our study, western blot and real-time PCR was used to detect CSP expression, which is more sensitive than immunogold electron microscopy in previous study. Therefore, it is reasonable for mutant oocysts to activate TEP1 at D5.

The results of real-time PCR and western blot detecting the CSP expression were uploaded as supplementary Figure 8. In addition to day 7, 9 and 12, we have also compared the CSP expression level between CSP_{mut} and CSP_{wt} oocysts in the infected mosquitoes, and still no significant difference was found. Therefore, the Figure 5a has been replaced with new one, please refer to the revised manuscript.

2. PPOs and oocyst killing. Several lines of evidence have implicated hemocyte derived PPOs in the mechanisms that mediate oocyst killing (Kwon and Smith, 2019-PNAS; Kwon et al 2020-bioRxiv), yet these are not discussed. This seems highly relevant since PPOs are precursors that are believed to initiate melanin biosynthesis, an integral component of the CSPmut phenotype presented by the authors. In addition, some thought should be placed into how the much earlier non-melanizing oocyst killing roles could later translate to melanization in the CSPmut background.

Response: Thanks. We totally agree with reviewer's point of view. According lines of evidence reported previously, melanization was directly mediated by PPOs, which is the secondary response of the death of parasite killed by TEP1. The question raised by reviewer is interesting to us, we therefore investigated whether PPOs were involved in this process. As a result, the mRNA levels of all PPO2, PPO3 and PPO9 were significantly down-regulated after TEP1 was knockdown, indicating that PPOs were participated in the melanization of oocysts killed by TEP1. This has been discussed now. Please refer to the revised manuscript.

-While parasite infections can be highly variable, I have concerns regarding the numbers of hemolymph sporozoites presented in the manuscript, which are in excess of 60,000 spz in many experiments, while in others range from 5-15,000. The latter number is much more in line with previously published studies.

Response: Thanks. We have checked the numbers of hemolymph sporozoites, and found no wrong with the number. However, the numbers were the total number of quantified mosquitoes, and not standardized to the number of sporozoite per mosquito. As the number of quantified mosquitoes was various between different experiments, then inconsistency appeared. We are very sorry for that. Now, the numbers of sporozoites and oocysts in figures have been standardized to per mosquito.

-The data presented in Figure 3 are not very robust. Based on morphology, the oenocytoids presented in Fig3b are not oenocytoids. Oenocytoids are rounded, without any extensions. This confounds the data presented in Figure 3b, where in addition both oenocytoid and prohemocyte are misspelled in the figure. I further doubt the data presented since the combined % of granulocytes and oenocytoids in Fig3b exceed 100%. In addition, >90% of cells following fixation should be granulocytes.

Response: Thanks. According to reviewer's suggestion, oenocytoids are recounted according to the morphology features of all three haemocytes, and the positive percentage of granulocyte or oenocytoid in all hemocytes has been recalculated. Our description of Y-axis title of Fig 3b might confuse reviewer. The percentage is calculated as the positive cells in granulocytes or oenocytoids, respectively, but not in all hemocytes. The word "granulocytes or oenocytoids" in the Figure has also been corrected. Fig.1 was now replaced by a new one.

In addition, the result of calculation of TEP1-expressing hemocytes based on morphology was confirmed FACS analysis, according to the method reported by Kwon, et al (2009, PNAS). The result was uploaded as supplementary Figure4.

-For the data presented in Figure 3D and Figure 4A, there is little description of the heatmaps presented. My interpretation is that only two replicates ("groups") are presented for each biological condition, and there is a great deal of variance between them. Furthermore, it would be beneficial to include gene abbreviations where possible for the individual genes included in the analysis to interpret the data without having to input every accession #. At present these figures provide little value.

Response: Thanks. Both Fig. 3d and 4a have been replaced by new ones, the genes of which have been indicated with corresponding gene ID and gene names. Hopefully, it is informative now.

-When paired with the issues of hemocyte descriptions in Figure 3b, the data presented in Fig4C are not very convincing that hemocytes undergo nitration. The DiI-labeling does not overlap very well in the images shown, and am not convinced that these are indeed hemocytes. Additional work is required to more accurately demonstrate that a significant portion of hemocyte undergo nitration by quantifying co-localization experiments with DiI and NY. Additionally, it may be interesting to follow-up with TEP1 and NY co-localization experiments as a reflection of immune activation. Are these immune activated or are they dead/dying in response to infection? It also would be interesting to know if there are potential differences in nitration between circulating and sessile (attached to midgut) hemocytes.

Response: Thanks. Additional experiment has been performed, and the Fig 4C has been replaced with a new one, which showed nitrated hemocytes around the mutant oocyst, but not to the WT oocyst. In addition, the colocalization of DiI and NY with TEP1 was also investigated, and it was found that about 50% of nitrated hemocytes also expressed TEP1, indicating that some of these cells are activated to express TEP1. In contrast, we did not find the significant expression of nitration in circulating hemocytes. As shown in Figure R1.

Figure R1. Representative immunofluorescent confocal image of circulating hemocytes from CSP_{wt} or CSP_{mut} parasites-infected mosquitoes at day 5 PI, stained with anti-nitrotyrosine antibody, DIL and DAPI.

-For the data presented in Figure 4C, do these reported hemocytes bind to the midgut in close proximity to oocysts? This seems relevant for their potential role in oocyst melanization and should be examined.

Response: Thanks. Additional experiment has been performed. In addition to around mutant oocysts as presented by Figure 4c, some hemocytes could be found to attach to the mutant oocysts, but not for WT oocysts, which has been uploaded as supplementary Figure 6.

-The in vitro hemocyte activation studies presented in Figure 5 are very difficult to interpret given the lack of experimental details provided. There is mention that midguts were treated with antibiotics, how was this done? How were the hemocytes collected? How many mosquitoes were used for each experiment to obtain hemocytes? If hemocytes were incubated with midgut homogenates, how were hemocytes identified? If I understand what was done, I would argue that these experiments are very artificial and may not accurately identify hemocytes from the rest of the homogenate. Furthermore, the article for which the methodology is derived from (Levashina et al 2001) does not mention a similar assay (they only use mosquito hemocyte-like cell lines) and wonder if this is the correct reference.

Response: Sorry for the lack of experimental details. Hemocytes were collected from about 20 naïve mosquitoes by perfusing method, and incubated with midgut homogenate from CSP_{wt}- and CSP_{mut}-infected mosquitoes in the presence of antibiotics. To identify the nitration of hemocytes, hemocytes were now stained with WGA (10 ug/mL, Invitrogen), which specifically marks mosquito hemocytes. Therefore, Figure 5c was replaced with new one. In addition, the reference of Levashina et al 2001 has been removed, as our method was not as same as that reported in the reference.

-I have several comments regarding the double infection experiments in Figure 6. First, evidence suggests that priming with a parasite reduces subsequent infections (Rodrigues et al 2010-Science), thus supporting the decrease RFP parasite infection intensity in those previously challenged mosquitoes. However, unlike Rodrigues et al. the parasites were allowed to persist

confounding the results. At the time of re-challenge, the initial infection (3 days post-infection) would not have any influence since CSP would not be expressed. As a result, any effects on RFP parasite melanization would be a “by-product” of persisting CSPmut parasites. Furthermore, the numbers of non-RFP parasites (CSPwt or CSPmut) should also be counted. Moreover, as oocysts are killed by melanization or other means, they lose their fluorescence, such that RFP parasites can no longer be distinguished from melanized CSPmut parasites. Therefore, a clear determination of the origin of the melanized parasites cannot be determined from the experiments displayed for Figure 6.

Response: Thanks. As demonstrated above, CSP_{mut} parasites are arrested at late oocyst stage, which is, to some extent, similar to what has been done in Rodrigues et al 2010-Science to switch the first-infected mosquitoes to 28°C to reduce oocyst survival. In addition, the expression of mutant CSP is gradually increased with the development of oocyst, and the activation of mosquito immune response by mutant CSP might persist for whole oocyst process. Therefore, the effect is resulted from persistent activation of mosquito immune response by mutant CSP.

The immune response elicited by the first infection of CSP_{mut} parasites mainly against oocyst stage of the second infection of 265BY-RFP. Although the second infection initiated 3 days post first infection, it will take 2 days for the second infected parasite to develop into early oocyst, at that time, the first-infected parasite has been developed for at least 5 days. We found the expression of CSP could be detected as early as 4-5 days by real-time PCR, and theoretically activate mosquito immune response against the oocyst of the second infected parasite. The reason to choose day 3 for the challenge has been explained in the result now. Please refer to the revised manuscript.

The melanized 265BY-RFP oocyst was counted under both fluorescent light and bright light through switching the light. The fluorescence of 265BY-RFP significantly fades away, when the oocyst was melanized. However, the detected residual fluorescence in highly melanized oocyst could help to identify its originating from 265BY-RFP, as indicated in our Fig.6b. Therefore, the origin of the melanized oocysts could be identified. However, we admitted the missing of some melanized 265BY-RFP oocysts due to the loss of fluorescence could not be completely excluded. Thus, the percentage of melanized 265BY-RFP oocysts is underestimated, but we think this will not affect our conclusion.

Minor comments

-Use of arrested to describe CSP mut throughout manuscript is somewhat misleading. Oocysts are formed, such that their development is not arrested, they develop normally. The phenotypes presented impair the ability of spz to ultimately reach the salivary gland. In my mind, this is an important distinction and can be misinterpreted. This terminology should be changed in the revised manuscript (line 101, 123, 156,277, 406,444,458)

Response: Thanks. The terminology “the developmental arrest of CSPmut oocysts” has been modified accordingly throughout manuscript. Please refer to the revised manuscript.

-Since mutations of both Py and PB are described in Figure 1, there is some confusion as to which species the remaining experiments are performed with. This should be transparent to the reader throughout the manuscript and would at least restate this in the figure legends.

Response: Thank. Plasmodium yoelii was used in the remaining experiments, which has been

described in the figure legends.

-I assume that infected mosquitoes are maintained on 10% sucrose after infection, but this is not directly stated. Please address.

Response: Yes. The infected mosquitoes are maintained on 10% sucrose after infection, which has been described in the Methods.

-The display of hemolymph spz is inconsistent throughout the manuscript, where numbers are displayed as scientific numbers or whole numbers.

Response: Thanks. To be consistent, all number has been standardized.

-For Figure 2C there is no mention in the figure legend what the white arrows are pointing to. The morphology of the developing oocysts seems different, but this really isn't expanded on. Furthermore, are we to assume that if an oocyst is melanized (Fig2a), then the internal sporozoites (Fig2c) are melanized as well?

Response: Thanks. The white arrows indicate the melanized or vacuolized sporozoites, and this information has been added in the figure legends of Figure 2. We found the melanization was firstly observed inside the oocyst, then gradually spread to the whole oocyst. This was also supported by our observation of TEP1 colocalization with CSP inside oocyst (Fig.2g). Therefore, we have rearranged the sequence of the result of Fig.2b and Fig.2c. Please refer to the revised manuscript.

-Was there any validation that the clodronate liposome treatment depletes phagocytic granulocytes in the proposed experiments?

Response: Yes. The depletion of phagocytic granulocytes by clodronate liposome was validated originally based on the morphology. Now, it has been confirmed by detected the relative expression of specific molecules, NimB2, LRIM15 and SCRB9 using real-time PCR. Please refer to the revised manuscript.

-For Figure 4d, adding example images of nitration differences in cells across different conditions would add to the rigor of these results.

Response: Thanks. In Fig.4d, the nitrification of midgut tissues from CSP_{mut}-infected mosquitoes with or without dsNOX5 was measured by alkaline phosphatase assay. We realized that we could not specify the nitration was resulted from hemocytes, because the nitration of other cells in midgut tissues could not be completely excluded. In addition, the similar experiment has been performed in Figure5e-f. Therefore, we think this result is not solid and necessary, and has been removed from the Figure 4. Please refer to the revised manuscript.

-For gene expression analysis throughout the manuscript, are these whole mosquitoes? Or specific tissues? Moreover, it also isn't always transparent what day of infection this analysis is being performed. This information should be clearly specified in the method.

Response: Thanks. Gene expression of whole mosquitoes was performed. This information, as well as the day of infection, have been clearly specified now.

-There is mention of vacuolation of oocysts in Figure 7, yet only briefly mentioned for Fig2 but

this isn't displayed in the manuscript. This should either be amended or removed.

Response: Thanks. The mention of vacuolation of oocysts in Figure 7 has been removed now.

-In the methods for the nitration assays, there is mention of using the nitrotyrosine antibody at a 1:100 dilution or a 1:3000 dilution. These are wildly different and if an antibody react at a 1:3000 dilution, using it at 1:100 will likely produce an artificial signal. Please address.

Response: Thanks for the suggestion. We checked our experiment record, 1:500 dilution of the anti-nitrotyrosine antibody was used, and it has been corrected. Sorry for the mistake.

Specific comments

-Line 78: While granulocytes have been implicated in the production of HdMV, this has not been conclusively proven. I suggest changing this phrasing to "hemocytes" .

Response: Thanks. The word "granulocytes" has been replaced with "hemocytes" now.

-Lines 82-84: Please modify to "PG stimulates the production of HDF" . Aside from a model proposed by Barletta et al 2019, there is no evidence that the components of HDF are actually produced in the fat body.

Response: Thanks. The sentence has been modified to "PG stimulates the production of HDF" .

-Lines 85-87: Please remove "a gene belonging to the susceptible Plasmodium falciparum strain". Mosquitoes are susceptible/resistant, not parasites.

Response: Thanks. "a gene belonging to the susceptible Plasmodium falciparum strain" has been changed into "a gene of Plasmodium falciparum" now.

-Figure 2d: The y-axis describes hemolymph spz, yet the text describes in vitro experiments. Please correct.

Response: Thanks. Hemolymph spz in y-axis has been corrected into "sporozoites released from oocysts *in vitro*".

-Figure 2G: Y axis is labeled "oocytes" instead of oocysts.

Response: Thanks. It has been corrected.

-Lines 195-196: While hemocytes undoubtedly produce TEP1, Volohonsky et al (2017 -Plos Pathogens) argue that TEP1 is predominantly produced in the fat body.

Response: Thanks. The statement has been modified as "Although a study argued that TEP1 was predominantly produced in the fat body, more evidence supported that hemocytes are regarded as the main source of TEP1".

-Lines 308-309: Delete "and induce oxidase stress" . A conformational change would promote immune detection first, oxidative stress could be a downstream response.

Response: Thanks. We are agree with the reviewer, and the phrase of "and induce oxidase stress" has been deleted.

-Lines 591 and 602: The name of the antibody is "nitrotyrosine" . Please correct.

Response: Thanks. The word of "nitrotyrosine" has been corrected.

Reviewer #3 (Remarks to the Author):

In this report, the authors show that when the conformation of the major sporozoite surface protein CSP is altered, the mosquito immune system is activated, resulting in death of oocysts. The authors provide evidence that activation of the immune system involves hemocyte nitration and upregulation of several genes in the Toll pathway, including the complement-like TEP1 gene.

- A major recommendation relates to rigor of data reporting. The authors state that several independent experiments have been conducted but present the results of only one “representative” (presumably the best one) experiment. I strongly feel that all data should be reported. That is, in each panel, the POOLED data of all the biological repeats should be reported, stating the actual number of biological repeats and the total number of entities (mosquitoes, guts, sporozoites, oocysts, etc.) that were assayed. This does NOT involve doing new experiments, but simply re-analyzing existing data.

Response: Thanks. The number of biological repeats and the total number of entities for each panel have been described, and the pooled data of all the biological repeats have been re-analyzed.

- No strong evidence is presented that the mutant parasite could be used for transmission-blocking vaccine and statements to this effect in the abstract, introduction and discussion should be omitted.

Response: Thanks. The statement of the potential role of mutant parasite for transmission-blocking vaccine in the abstract, introduction and discussion has been deleted.

- A comment is needed to explain how CSP with a defective pexel domain reaches the surface of the oocyst (Fig. 2g) and of the sporozoite (Fig. 5b).

Response: Thanks. We postulated that the mutation of pexel domain affected the integrity of the inner membrane complex of the parasite and this potentially allowing recognition of the oocyst by the circulated hemocytes, instead of the move of mutant CSP to the surface of the oocyst. The possible mechanism of mutant CSP of oocyst and sporozoite to be sensed by mosquito immunity has been discussed.

- In some experiments, very large number of mutant sporozoites are found in the hemolymph (e.g., Fig. 3c, 4g). Issues that need to be addressed: 1) with what efficiency (if any) do these sporozoites invade the salivary glands? 2) are these sporozoites (either from salivary glands or from hemolymph) infective to mice? 3) is the motility of the mutant sporozoites affected? Data addressing these questions would significantly enhance the value of the findings.

Response: Thanks. We didn't find sporozoites in salivary gland, although hemolymph sporozoites could be detected after granulocytes were depleted or NOX5 was silenced in Fig. 3c and Fig.4g. This could be explained as mutant sporozoites in hemolymph were cleared by mosquito immune responses prior to reaching the salivary gland, or they lost the ability to invade salivary gland. As the number of the re-appeared hemolymph sporozoites is still much less than that of WT, we

prefer the possibility that mutant sporozoites released into hemolymph were cleared by mosquito immunity.

We have recorded the movement of both mutant and WT sporozoites by video, and found that the motility of mutant sporozoites was comparable to that of WT sporozoites. As for the infectivity of hemolymph sporozoites to mice, we found 40% of mice could be infected by CSP_{WT} hemolymph sporozoites, but none mice were infected when injected with CSP_{mut} hemolymph sporozoites (Table R2).

Our study (Frontiers in Immunology, 2022, accepted, DOI: 10.3389/fimmu.2022.815936) has demonstrated that the pexel domain is essential for *P.berghei* CSP to resist the killing effect of IFN- γ *in vivo*. Therefore, this phenomenon may be explained by the loss of resistance of *P.yoelii* CSP_{mut} hemolymph sporozoites to IFN- γ *in vivo*. To exclude the effect of IFN- γ on the infectivity of hemolymph sporozoites, the infection of hepatocytes with both hemolymph sporozoites *in vitro* was performed. As a result, similar infectivity of WT and mutant hemolymph sporozoites to hepatocytes were observed, as their parasite burden in HepG2-CD81 determined by real-time PCR was comparable at 42 h post incubation *in vitro* (Supplementary Figure2). Taken together, we think both the motility and infectivity of the mutant sporozoites was not significantly affected. The video and Figure have now been uploaded as supplemental materials. Please refer to the revised manuscript.

Table R2. The infectivity of Hemolymph sporozoites in mice

Parasite	Sporozoites	Infected mice	Positive mice	Positive rate (%)
CSP _{WT}	1 x 10 ⁶	5	3	60
	5 x 10 ⁵	5	2	40
	1 x 10 ⁵	5	1	20
CSP _{mut}	1 x 10 ⁶	5	0	0
	5 x 10 ⁵	5	0	0
	1 x 10 ⁵	5	0	0

- The Western blot of Fig. 2f uses ‘S7’ as “an internal control”. This is puzzling, as S7 is a ribosomal protein. How can it be found in the hemolymph in large amounts and used as a reference? What is the source of anti-S7 antibody?

Response: Thanks. Hemolymph, collected by PBS perfusion, also containing hemocytes, which were collected by centrifugation. S7 should come from hemocytes. Anti-S7 polyclonal antibody was prepared by company, which has been described in Methods.

- An apparent inconsistency: in Fig. 1g no hemolymph sporozoites were detected, while in Fig. 2i several thousand sporozoites were detected.

Response: Thanks. Actually, both hemolymph and salivary gland sporozoites are developed in the mosquitoes, but the number is too low to be seen, as compared to the WT sporozoites. Fig 1 has been modified to show the low number of both hemolymph and salivary gland sporozoite.

- Nature Communications has a broad readership and articles should be easily accessible to the average reader. Figures 3d and 4a are not easy to understand. Perhaps a more extensive legend might help.

Response: Thanks. The legends of Figures 3d and 4a have been extended for readers to understand.

- Supplementary Table 1 needs a legend. What are the percentages? Are the values relative to dsGFP?

Response: Yes. Now, the legend for Supplementary Table 1 has been added.

- Fig. 3b and Methods. What criteria were used to distinguish granulocytes from oenocytes?

Response: Thanks. The identification of hemocytes was originally based on morphology. Now, it has been confirmed by detected the relative expression of specific molecules, NimB2, LRIM15 and SCRB9, using real-time PCR. Please refer to the revised manuscript.

- Fig. 3f and 3g. The legend says “... in WT or mutant parasite infected mosquitoes”. Are the data shown for WT or mutant? One of the two appears to be missing.

Response: Sorry for the mistake. Only mutant parasite, but not WT, was shown. The legend has been corrected now.

- Legend of Fig. 4d. Explain how measurements were done.

Response: In Fig.4d, the nitrification of midgut tissues from CSP_{mut}-infected mosquitoes with or without dsNOX5 was measured by alkaline phosphatase assay. We realized that we could not specify the nitration was resulted from hemocytes, because the nitration of other cells in midgut tissues could not be completely excluded. In addition, the similar experiment has been performed in Figure5e-f. Therefore, we think this result is not solid and necessary, and has been removed from the Figure 4. Please refer to the revised manuscript.

- Fig. 5a and Methods. What is the origin of the anti-HSP70 antibody?

Response: Thanks, anti-HSP70 was prepared by company, which has been described in Methods now.

- Figs. 6a, 6d 6f and 6h: what is “KM” ?

Response: “KM” means Kunming mice. As the information is not necessary, “KM” has been removed from the Figure.

- Figs. 6b and 6e. I could not understand these figures. Are the oocyst numbers a sum of those originating from the first and the second infection? Can fluorescence be detected in melanized oocysts? Three curves are apparently plotted but only two are visible.

Response: The oocyst number is originated from the second infection with 265BY-RFP. The total number of 265BY-RFP was counted under fluorescence microscope, and the melanized 265BY-RFP oocyst was counted under both fluorescent light and bright light through switching the light. The fluorescence of 265BY-RFP significantly fades away, when the oocyst was melanized. However, the detected residual fluorescence in highly melanized oocyst could help to identified

its originating from 265BY-RFP, as indicated in our Fig.6b. Therefore, the origin of the melanized oocysts could be identified. However, we admitted the missing of some melanized 265BY-RFP oocysts due to the loss of fluorescence could not be completely excluded.

- Fig. 6d. After spending much time trying to understand, it appears that the label is wrong: middle black bar should be labeled CSPWT+RFP, not CSPmut+RFP, is this correct?

Response: Sorry for the confusion. The label in Fig.6d is wrong, which has been corrected now.

- Fig. 6e: Three curves are apparently plotted but only two are visible. Is dsNOX5 100% effective in suppressing melanization (cf. Fig. 5e)?

Response: Yes. dsNOX5 can totally suppress melanization, and line labels have been modified to make all three curves to be visible. In revised Figure 6, the order of Fig.6e and 6f has been exchanged now.

- Fig. 6e: it is confusing; the figure says “percentage of melanized oocysts/mature oocysts (%)” and the legend says “average number of P.y BY265-RFP oocysts” .

Response: Sorry for the confusion of the Fig. 6e and Fig. 6f legends. In revised Figure 6, the order of Fig.6e and 6f has been exchanged now. Legend of Fig.6e should be “average number of P.y BY265-RFP oocysts” .

- Fig. 6f: it is confusing; the figure says “oocyst number (RFP)” and the legend says “percentage of melanized oocysts”

Response: Sorry for the confusion of the Fig. 6e and Fig. 6f legends. In revised Figure 6, the order of Fig.6e and 6f has been exchanged now. Legend of Fig.6f should be “percentage of melanized oocysts/mature oocysts (%)” .

- Fig. 7. A key component of the proposed model is that TEP1 is the effector that causes oocyst death. This assumption is easily testable but knockdown of TEP1 experiments have not been conducted. This is a significant weakness.

Response: Thanks. Actually, we have performed the TEP1 knockdown experiment, and found knockdown of TEP1 could completely revert the phenotype of CSPmut oocyst melanization. This result was presented in Figure 2h-i.

- Improvement of the English writing may be required.

Response: Thanks. The English writing of the revised manuscript has been improved by Editage.

In summary, the manuscript presents interesting experiments that report on a novel way that mosquito immunity is activated. However, weaknesses are noted and the lack of rigor in assembling the manuscript is disappointing, to say the least.

Response: Thanks a lot for reviewer’s comments, which will help us to significantly improve our manuscript. Hopefully, reviewers will satisfy with our responses.

Reviewers' Comments:

Reviewer #1:

Remarks to the Author:

The authors have addressed most of the issues that were raised. I still have some comments that should be addressed before acceptance.

Arrows in Fig. 4c are not explained.

Fig. 5b. In the reference Coppi et al the label of anti-C and anti-N CSP antibodies is slightly different from what is reported here, is there an explanation? A comment should be added to address this.

Line 427: according to Fig. 6h after NOX5 knockdown a few salivary gland sporozoites appear (estimated <500 compared to 7000 in the control). The sentence should be modified to acknowledge that this is not a complete rescue.

Line 461 replace «were» with «are», add a reference

Line 478, the experiments were carried out in two very similar rodent parasites, and before analysis of the distantly related *P. falciparum* it is pre-mature to state that the mechanism described here is «common».

Supplementary videos. I am not sure these videos show motile sporozoites. In my opinion they are not attached to the surface. A better assay is to allow gliding on glass slides and labeling with anti-CSP to visualize the trails. Also, there is no explanation for the videos, which is WT and mutant? Line 176 refers to Supplementary video 1 and Supplementary Fig. 1. Please correct and explain.

The complete dataset from the transcriptomics analysis should be included as supplementary information. The data should also be submitted to a suitable data repository and the link provided.

Reviewer #2:

Remarks to the Author:

The current submission is much improved and appreciate the time and effort that the authors have put into the revised manuscript. However, with the new data and revised text in the revised manuscript, I still have some concerns regarding the interpretations of the data that need to be resolved. In addition, there are some additional considerations of parasite killing that have yet to be fully addressed, where a more thorough review of the literature is needed to place the outcomes of this study in the context of previously published work. My comments are as follows:

Major comments

-While the "late-phase" immune responses that limit oocyst survival are briefly mentioned, this is not adequately covered in the introduction (Lines 90-93) or in the discussion (Lines 460-463). Not only do immune responses that limit the oocyst stage involve mosquito hemocytes (PMID: 26080400, 31235594, 34484178), an important aspect in the context of this study, these responses are mediated by PPOs (PMID: 31235594, 34484178) and are independent of TEP1 function (PMID: 26080400, 28764765). These studies are particularly important to provide accurate context to the results of this study.

-Similar to the above comment, previously described "late-phase" responses occur prior to the development of the mature oocyst, believed to occur sometime between day 2 and day 8 post-infection (PMID: 26080400, 31235594, 34484178), and do not appear to directly involve melanization of the oocyst (PMID: 31235594, 34484178). As a result, I would argue that the recognition of the mature oocyst and involvement of TEP1 as described in this study represents an additional mechanism by which oocysts can be recognized and targeted in the mosquito host. This should be discussed in detail in a revised discussion to properly and fully place the findings of this study in the context of the previous work examining oocyst killing responses.

-In addition to the above comment, an alternative explanation could be differences in the immune response between *An. stephensi* (this study) and previous studies performed in *An. gambiae*. This

should also be discussed in a revised manuscript.

-While I commend the efforts to perform FACS on mosquito hemocyte populations (Figure S4), it isn't clear how the cell types were distinguished using only WGA and DRAQ5. The authors cite Kwon et al. - 2019 which used a similar approach, but this study did not define the immune cell populations using gating alone. Phagocytic granulocytes were distinguished in this study by the phagocytosis of fluorescent beads, while the other cell types were not identified. While ploidy has been seen for mosquito hemocytes via FACS (PMID: 24363411, 26515540, 31235594, 34318744) as suggested by the different groups distinguished by DRAQ5 intensity in Figure S4, this is not an established method to distinguish mosquito hemocyte subtypes. Ploidy was used to gate cell populations, that ultimately did see some enrichment of specific cell types (PMID: 34318744), yet this was not absolute. If anything, this previous study also suggests that those cells with the highest ploidy were oenocytoids, not granulocytes as the authors suggest in Figure S4. As a result, the authors should temper this type of analysis shown in Figure S4b. I think the data presented in Figure S4c can still be used to solidify this argument that hemocytes display increased TEP1 expression, however, these data need to be reanalyzed combining the three cell types since they cannot be accurately distinguished. As such these results and corresponding text should be modified.

-After a closer look at the RNA-seq studies in Figure 3D and Figure 4A, I think these should be moved to the supplement. I appreciate the improved annotations and think there is still some value in these data, but this analysis consists of only 2 replicates and am not sure how the authors are able to perform any kind of robust statistical analysis on these data. From the heat maps alone, the data look highly variable. As a result, the authors should not make claims of "significance" in the results text.

-The authors need to provide the RNAseq raw and processed data. The link provided to the raw data in the M&M is for Kwon et al. 2019, a completely independent study not performed by the authors! In addition, Supplementary Table 4 does not provide sufficient detail of gene expression levels, statistical analysis, etc. Also, there is no mention of genes that are down-regulated.

Minor comments

-For Figure S3, it appears as though only PPO2, PPO3, and PPO9 are examined. Since there are nine PPOs in Anopheles, it is not clear why only these were examined. I suspect this is because the aforementioned PPOs have been implicated in oocyst killing responses (PMID: 31235594, 34484178), but this should be clearly stated. Also, if this is the case, it is unclear why PPO1 (PMID: 34484178) was excluded from the analysis.

Specific comments

-Line 165-167: Rephrase. I suggest the following: "Since TEP1 is an important component of malaria parasite killing (insert references), we investigated whether..."

-Line 220-222: There are more specific references to suggest that TEP1 is expressed in hemocytes (PMID: 27624304, 34318744)

-Line 228: Please correct and remove oenocytoids. Previous studies (PMID: 31235594, 33604338, 34484178) and the data presented in Figure S5 suggest that only granulocytes are depleted following clodronate treatment.

-In heatmaps in Figure 3D and Figure 4A, "trendancy" should be changed to "trend"

-Lines 301-304: Please modify this sentence to increase clarity.

-Lines 405-407: Please modify this sentence to increase clarity.

-Lines 415-416: Change "identified as residual fluorescence remaining" to "identified by residual fluorescence"

-Lines 460-463: Previous studies have suggested ~50% loss in parasite numbers between the early and mature oocyst stages (PMID: 19454353, 26080400, 28764765, 31235594, 34484178). Evidence suggests that this response is mediated by mosquito hemocytes (PMID: 26080400, 31235594, 34484178) and involves PPOs (PMID: 31235594, 34484178) independent of TEP1 function (PMID: 26080400, 28764765).

-Lines 467-468: The following citations should be also included PMID: 31235594, 33789941, 34484178

Reviewer #3:

Remarks to the Author:

The revised manuscript is substantially improved.

A few minor comments:

- Legend to Fig 2i. It states "in mosquitoes (n = 60) infected with CSPwt or CSPmut parasites..." but data for only one parasite (presumably CSPmut) is shown. This needs to be corrected.
- Lines 487-488 state "we postulated that the conformational change in CSP might change the physical surface of oocysts, which would be sensed by the circulating hemocytes". Given that oocysts are separated from the hemolymph by the basal lamina, how is this physical interaction postulated?
- The manuscript could further be improved by proofing the English.

Dear reviewers,

Thanks a lot for your comments, which will further improve our manuscript. We have responded to each comment raised by reviewers point by point as following. Hope you will be satisfied with the replies and the modification of the manuscript. Please see the revised manuscript and supplementary materials, and the changes in the manuscript text have been highlighted by yellow color.

REVIEWER COMMENTS

Reviewer #1 (Remarks to the Author):

The authors have addressed most of the issues that were raised. I still have some comments that should be addressed before acceptance.

Arrows in Fig. 4c are not explained.

Response: Thanks for the reminder. Now, the arrows in Fig.4c have been explained in the figure legends.

Fig. 5b. In the reference Coppi et al the label of anti-C and anti-N CSP antibodies is slightly different from what is reported here, is there an explanation? A comment should be added to address this.

Response: Thanks. We noted a slightly different pattern of hemolymph sporozoites labelled by anti-C-terminal CSP polyclonal antibody between ours and previous study. It might be explained as a small peptide was used for the preparation of the anti-sera in our experiment, and the full length of TSR region in C-terminus of CSP were used in previous study. Now, the explanation has been added in the revised manuscript.

Line 427: according to Fig. 6h after NOX5 knockdown a few salivary gland sporozoites appear (estimated <500 compared to 7000 in the control). The sentence should be modified to acknowledge that this is not a complete rescue.

Response: Thanks. The words of “a few of” has been added before “infectious salivary gland sporozoites appeared”.

Line 461 replace «were» with «are», add a reference

Response: Thanks. The word “are” has been replace by “were”. The reference of Trop Med Int Health in 1998 has been added.

Line 478, the experiments were carried out in two very similar rodent parasites, and before analysis of the distantly related *P. falciparum* it is pre-mature to state that the mechanism described here is «common».

Response: Thanks. The common has been deleted now.

Supplementary videos. I am not sure these videos show motile sporozoites. In my opinion they are not attached to the surface. A better assay is to allow gliding on glass slides and labeling with anti-CSP to visualize the trails. Also, there is no explanation for the videos, which is WT and mutant?

Response: Thanks for the suggestion. The assay has been re-performed according to your suggestion. Please refer to the up-loaded videos.

Line 176 refers to Supplementary video 1 and Supplementary Fig. 1. Please correct and explain.

Response: Thanks. In line 176, the comparable infectivity of hemolymph sporozoites between CSP_{mut} and CSP_{wt} is supported by the result of Supplementary Fig.2, not Supplementary Fig.1. Are we missing something?

The complete dataset from the transcriptomics analysis should be included as supplementary information. The data should also be submitted to a suitable data repository and the link provided.

Response: Thanks. The complete dataset from the transcriptomics analysis has been uploaded as Supplementary information, and the related link has been provided.

Reviewer #2 (Remarks to the Author):

The current submission is much improved and appreciate the time and effort that the authors have put into the revised manuscript. However, with the new data and revised text in the revised manuscript, I still have some concerns regarding the interpretations of the data that need to be resolved. In addition, there are some additional considerations of parasite killing that have yet to be fully addressed, where a more thorough review of the literature is needed to place the outcomes of this study in the context of previously published work. My comments are as follows:

Major comments

-While the “late-phase” immune responses that limit oocyst survival are briefly mentioned, this is not adequately covered in the introduction (Lines 90-93) or in the discussion (Lines 460-463). Not only do immune responses that limit the oocyst stage involve mosquito hemocytes (PMID: 26080400, 31235594, 34484178), an important aspect in the context of this study, these responses are mediated by PPOs (PMID: 31235594, 34484178) and are independent of TEP1 function (PMID: 26080400, 28764765). These studies are particularly important to provide accurate context to the results of this study.

Response: Thanks a lot for the suggestion. We have checked all the references you provided, and have a deeper understanding about this field. The sentences in both introduction and discussion have been re-written. Please refer to the revised manuscript.

-Similar to the above comment, previously described “late-phase” responses occur prior to the development of the mature oocyst, believed to occur sometime between day 2 and day 8 post-infection (PMID: 26080400, 31235594, 34484178), and do not appear to directly involve melanization of the oocyst (PMID: 31235594, 34484178). As a result, I would argue that the recognition of the mature oocyst and involvement of TEP1 as described in this study represents an additional mechanism by which oocysts can be recognized and targeted in the mosquito host. This should be discussed in detail in a revised discussion to properly and fully place the findings of this study in the context of the previous work examining oocyst killing responses.

Response: Thanks a lot for the suggestion. We agree with your opinion, and realize the novelty and difference of our finding as compared to previous work, which has been discussed in the last paragraph of our revised manuscript.

-In addition to the above comment, an alternative explanation could be differences in the immune response between *An. stephensi* (this study) and previous studies performed in *An. gambiae*. This should also be discussed in a revised manuscript.

Response: Thanks. The alternative explanation has been added in the last paragraph of the discussion.

-While I commend the efforts to perform FACS on mosquito hemocyte populations (Figure S4), it isn't clear how the cell types were distinguished using only WGA and DRAQ5. The authors cite Kwon et al. -2019 which used a similar approach, but this study did not define the immune cell populations using gating alone. Phagocytic granulocytes were distinguished in this study by the phagocytosis of fluorescent beads, while the other cell types were not identified. While ploidy has been seen for mosquito hemocytes via FACS (PMID: 24363411, 26515540, 31235594, 34318744) as suggested by the different groups distinguished by DRAQ5 intensity in Figure S4, this is not an established method to distinguish mosquito hemocyte subtypes. Ploidy was used to gate cell populations, that ultimately did see some enrichment of specific cell types (PMID: 34318744), yet this was not absolute. If anything, this previous study also suggests that those cells with the highest ploidy were oenocytoids, not granulocytes as the authors suggest in Figure S4. As a result, the authors should temper this type of analysis shown in Figure S4b. I think the data presented in Figure S4c can still be used to solidify this argument that hemocytes display increased TEP1 expression, however, these data need to be reanalyzed combining the three cell types since they cannot be accurately distinguished. As such these results and corresponding text should be modified.

Response: Thanks a lot for your suggestion. We agree that the subsets of hemocytes, especially for oenocytoids and granulocytes, cannot be accurately distinguished by DRAQ5 intensity via FACS. Therefore, we reanalyzed the data of Figure S4a-c by the combination of the oenocytoids and granulocytes. In addition, the upregulation of TEP1 in granulocytes has also been confirmed by the phagocytosis of fluorescent beads, which has been uploaded as Figure S4d-f. In conclusion, the upregulation of TEP1 in granulocytes has been strongly supported by our data, but the upregulation of TEP1 in oenocytoids couldn't be excluded. Thus, the results and corresponding text have been modified accordingly.

-After a closer look at the RNA-seq studies in Figure 3D and Figure 4A, I think these should be moved to the supplement. I appreciate the improved annotations and think there is still some value in these data, but this analysis consists of only 2 replicates and am not sure how the authors are able to perform any kind of robust statistical analysis on these data. From the heat maps alone, the data look highly variable. As a result, the authors should not make claims of "significance" in the results text.

Response: Thanks. We are agreeing with you, and have moved both Figure 3D and Figure 4A, as well as the corresponding methods, into the supplementary materials. The statistical analysis of 2 replicates of our RNA-seq data could not be performed, and it can only show the trend of the upregulation of both Toll-pathway and antioxidant related genes. The word "significance" in the result text now has been deleted.

-The authors need to provide the RNAseq raw and processed data. The link provided to the raw data in the M&M is for Kwon et al. 2019, a completely independent study not performed by the authors! In addition, Supplementary Table 4 does not provide sufficient detail of gene expression levels, statistical analysis, etc. Also, there is no mention of genes that are down-regulated.

Response: We are very sorry for the mistake. It has been corrected. In addition, the details about the levels of genes down-regulated or up-regulated have been added in Supplementary Table 4. As the statistical analysis could not be performed, the upregulation trend of pathway instead was shown.

Minor comments

-For Figure S3, it appears as though only PPO2, PPO3, and PPO9 are examined. Since there are nine PPOs in Anopheles, it is not clear why only these were examined. I suspect this is because the aforementioned PPOs have been implicated in oocyst killing responses (PMID: 31235594, 34484178), but this should be clearly stated. Also, if this is the case, it is unclear why PPO1 (PMID: 34484178) was excluded from the analysis.

Response: Sorry for the missing of PPO1. Now, the reason for us to evaluate the role PPOs in the melanization of mutant oocysts has been stated, and the data of PPO1 was also added in FigureS3.

Specific comments

-Line 165-167: Rephrase. I suggest the following: “Since TEP1 is an important component of malaria parasite killing (insert references), we investigated whether…”

Response: Thanks. The sentence has been rephrased as what you suggested.

-Line 220-222: There are more specific references to suggest that TEP1 is expressed in hemocytes (PMID: 27624304, 34318744)

Response: Thanks. The references have been replaced by the more specific ones as you suggested.

-Line 228: Please correct and remove oenocytoids. Previous studies (PMID: 31235594, 33604338, 34484178) and the data presented in Figure S5 suggest that only granulocytes are depleted following clodronate treatment.

Response: Thanks. Oenocytoids has been removed.

-In heatmaps in Figure 3D and Figure 4A, “trendancy” should be changed to “trend”

Response: Thanks. It has been changed to “trend of pathway”.

-Lines 301-304: Please modify this sentence to increase clarity.

Response: Thanks. The expression has been modified. We hope it is clearer now.

-Lines 405-407: Please modify this sentence to increase clarity.

Response: Thanks. The expression has been modified. We hope it is clearer now.

-Lines 415-416: Change “identified as residual fluorescence remaining” to “identified by

residual fluorescence”

Response: Thanks. It has been corrected.

-Lines 460-463: Previous studies have suggested ~50% loss in parasite numbers between the early and mature oocyst stages (PMID: 19454353, 26080400, 28764765, 31235594, 34484178). Evidence suggests that this response is mediated by mosquito hemocytes (PMID: 26080400, 31235594, 34484178) and involves PPOs (PMID: 31235594, 34484178) independent of TEP1 function (PMID: 26080400, 28764765).

Response: Thanks a lot for the suggestion. In logically, we think it is better to add this sentence in the last paragraph of discussion to highlight the difference of our finding and its explanation, and it has been added in the context of last paragraph. We hope you will be satisfied with our revision.

-Lines 467-468: The following citations should be also included PMID: 31235594, 33789941, 34484178

Response: Thanks. The citations have been added.

Reviewer #3 (Remarks to the Author):

The revised manuscript is substantially improved.

A few minor comments:

- Legend to Fig 2i. It states “in mosquitoes (n = 60) infected with CSPwt or CSPmut parasites...” but data for only one parasite (presumably CSPmut) is shown. This needs to be corrected.

Response: Thanks. It has been corrected by removing words “CSPwt or”.

- Lines 487-488 state “we postulated that the conformational change in CSP might change the physical surface of oocysts, which would be sensed by the circulating hemocytes”. Given that oocysts are separated from the hemolymph by the basal lamina, how is this physical interaction postulated?

Response: Thanks. I don't know how did mutant oocyst activate hemocytes. Our postulation of the physical interaction between mutant oocyst and hemocytes was based on the finding of the attachment of hemocytes to mutant oocysts, which is presented in the supplementary Figure 8. This has been interpreted in the discussion, please refer to our revised manuscript.

- The manuscript could further be improved by proofing the English.

Response: Thanks. The revised manuscript has been sent to Editage (www.editage.cn) for English language editing.

Reviewers' Comments:

Reviewer #1:

Remarks to the Author:

The authors have answered most of my comments satisfactorily and edited the manuscript accordingly.

However, regarding the videos of moving sporozoites. 1) There are two videos, but there is no mentioning of video number 2 - please address

2)The method used for the imaging is not the standard which is allowing sporozoites to glide on a coated slide and labeling the trails of CSP that are deposited. However, the videos are acceptable in their present form.

Reviewer #2:

Remarks to the Author:

I have no further comments. I think the authors have done an excellent job addressing my previous concerns.

REVIEWERS' COMMENTS

Reviewer #1 (Remarks to the Author):

The authors have answered most of my comments satisfactorily and edited the manuscript accordingly.

However, regarding the videos of moving sporozoites. 1) There are two videos, but there is no mentioning of video number 2 - please address

Response: Sorry for the mistake. Now, both video 1 and video 2 have been mentioned in the main text. As suggested by editor, the 'Video' has also been changed into 'Movie', and the text of ' those of the CSP_{wt} hemolymph sporozoites *in vitro* (Supplementary Video 1 & Supplementary Fig. 2)' has also been changed into 'those of the CSP_{wt} hemolymph sporozoites *in vitro* (Supplementary Movie 1,2 & Supplementary Fig. 2)'. Please refer to the revised manuscript.

2)The method used for the imaging is not the standard which is allowing sporozoites to glide on a coated slide and labeling the trails of CSP that are deposited. However, the videos are acceptable in their present form.

Response: Thanks a lot for your suggestion which will improve our work in the future. We will evaluate the motility of sporozoites using the standard method, as you suggested, in our future study.

Reviewer #2 (Remarks to the Author):

I have no further comments. I think the authors have done an excellent job addressing my previous concerns.

Response: Thanks. We are appreciated for your comments, which have greatly improved our manuscript.